# Functional locus coeruleus imaging to investigate an ageing noradrenergic system

Mareike Ludwig [1,2,3] ✉, Yeo-Jin Yi[1,3], Falk Lüsebrink[3,4,5], Martina F. Callaghan [6], Matthew J. Betts[1,2,3], Renat Yakupov [3], Nikolaus Weiskopf [6,7,8], Raymond J. Dolan [6,9], Emrah Düzel[1,3,10,12] & Dorothea Hämmerer [1,2,3,6,10,11,12]

The locus coeruleus (LC), our main source of norepinephrine (NE) in the brain, declines with age and is a potential epicentre of protein pathologies in neurodegenerative diseases (ND). In vivo measurements of LC integrity and function are potentially important biomarkers for healthy ageing and early ND onset. In the present study, high-resolution functional MRI (fMRI), a reversal reinforcement learning task, and dedicated post-processing approaches were used to visualise age differences in LC function ($N = 50$). Increased LC responses were observed during emotionally and task-related salient events, with subsequent accelerations and decelerations in reaction times, respectively, indicating context-specific adaptive engagement of the LC. Moreover, older adults exhibited increased LC activation compared to younger adults, indicating possible compensatory overactivation of a structurally declining LC in ageing. Our study shows that assessment of LC function is a promising biomarker of cognitive aging.

Ageing is a multifactorial process, often accompanied by cognitive decline such as memory impairments, which can affect the ability to live a self-determined, independent life. Post-mortem work shows that, among brain structures affected in ageing, neuromodulatory nuclei of the brainstem, such as the noradrenergic locus coeruleus (LC), appear to be particularly affected by age-related neurophysiological decline[1,2]. The LC is our primary source of norepinephrine (NE)[3,4] in the brainstem and projects to various brain structures, including the hippocampus (HPC), amygdala, thalamus and parietal cortex, thereby serving as a critical modulator of memory, perception, attention, and learning[5–10]. Post-mortem data indicate that tau pathology, a typical feature of neurodegenerative diseases (ND) like Alzheimer´s disease (AD), can be detected in the LC during the earliest stages of AD before it spreads to the cortex and prior to the onset of clinically noticeable cognitive symptoms[11–13]. Concurrently with non-pathological ageing, the mean LC signal intensity values (contrast ratios, CRs) decline in the rostral parts[14]. This leads to decreased noradrenergic signalling in the

brain[12], a change that has been associated with a decline in cognitive function, attention, and memory in the elderly[8]. While the LC-NE system may be particularly vulnerable, research also suggests that a structurally intact LC is a better predictor of a more advantageous cognitive development in ageing as compared to more intact serotonergic or dopaminergic nuclei[15]. Indeed, several studies have shown that ageing individuals with more structurally intact LC show less cognitive decline[14,16–19]. Mounting evidence thus suggests that LC integrity and function serve as important biomarkers of both healthy aging and early onset of ND (for review see Engels-Domínguez et al.[20]). Given that neuronal function loss likely occurs before cell death[21], it is crucial to also consider functional indicators of LC decline in ageing. However, functional assessments of the LC are methodologically challenging in humans due to its very small size (about 1-3 mm wide and 15 mm long, see Fernandes et al.[22]), which requires dedicated imaging sequences and, in particular, dedicated data analysis techniques[23]. The aim of this study was to investigate age differences in functional activations of the LC. Therefore, we

¹Institute of Cognitive Neurology and Dementia Research, Otto-von-Guericke University Magdeburg, Magdeburg, Germany. ²CBBS Center for Behavioral Brain Sciences, Magdeburg, Germany. ³German Center for Neurodegenerative Diseases (DZNE), Magdeburg, Germany. ⁴Biomedical Magnetic Resonance, Faculty of Natural Sciences, Otto-von-Guericke University, Magdeburg, Germany. ⁵NMR Methods Development Group, Max Planck Institute for Human Cognitive and Brain Sciences, Leipzig, Germany. ⁶Wellcome Centre for Human Neuroimaging, UCL Queen Square, Institute of Neurology, University College London, London, UK. ⁷Department of Neurophysics, Max Planck Institute for Human Cognitive and Brain Sciences, Leipzig, Germany. ⁸Felix Bloch Institute for Solid State Physics, Faculty of Physics and Earth Sciences, Leipzig University, Leipzig, Germany. ⁹Max Planck University College London Centre for Computational Psychiatry and Ageing Research, London, UK. ¹⁰Institute of Cognitive Neuroscience, University College London, London, UK. ¹¹Department of Psychology, University of Innsbruck, Innsbruck, Austria. ¹²These authors contributed equally: Emrah Düzel, Dorothea Hämmerer. ✉e-mail: mareike.ludwig@med.ovgu.de

**Table 1 | Four types of event-related GLMs with corresponding regressors as well as contrasts of interest**

| Types of event-related GLMs | regressors | contrast of interest |
|---|---|---|
| (1) emotional salience | forced choice, free choice, reversal, gain, loss, fixation, response left, response right | loss feedback > gain feedback |
| (2) task-related salience | forced choice, free choice, reversal, no reversal, loss, fixation, response left, response right | reversal feedback > no reversal feedback |
| (3) memory performance | forced choice, reversal, gain, loss, response left, response right, remembered, not remembered | remembered > not remembered |
| (4) emotional memory performance | forced choice, reversal, gain, loss, response left, response right, remembered loss, remembered gain, not remembered before loss, not remembered before gain | remembered before loss feedback > not remembered before loss feedback |

used high-resolution functional magnetic resonance imaging (fMRI) (1.5 mm) in conjunction with a newly developed MR data analysis pipeline, which allows for sufficient spatial precision in analyses of brainstem activations via a rigorous post-processing procedure[24] (for review see Liu et al.[25]). To engage the LC, we developed a reversal reinforcement learning task which included events of emotional and task-related salience as well as differential memory effects of such events. The motivation for this task arose from electrophysiological studies in rodents and monkeys, as well as physiological studies in humans, suggesting that the LC-NE system is generally involved in the processing and encoding of salient events such as negative emotional events[1,17,26] or identifying change points in reward association learning[17,27,28]. Animal studies have shown that NE release from the LC during such tasks supports memory encoding in the HPC via ß-adrenoceptors[29]. In line with this, human studies have shown higher activation in the LC during the encoding of emotionally salient events[30] as well as a better memory for emotionally salient events in individuals with higher LC integrity[17,31,32]. In summary, although there is increasing evidence that structural decline of the LC with age can be demonstrated in vivo, there is little evidence of age differences in LC function, which presumably should be associated with structural decline. In our study, we wanted to build upon these findings by assessing whether (1) (emotionally) salient events such as negative feedback (loss feedback > gain feedback) are associated with increased LC activation, (2) whether task-related salient events, such as condition reversals (reversal feedback > no reversal feedback), are associated with increased LC activation, and whether (3-4) LC activation during such events contributes to memory performance (remembered > not remembered) for salient events (remembered before loss feedback > not remembered before loss feedback; see Table 1). Moreover, by comparing younger and older adults, we investigated (5) age differences in LC reactivity in these instances.

**Behavioural results**

Regarding memory effects of (1) emotional salience, a statistically significant interaction between trials before vs. trials after feedback and loss vs. gain feedback was observed, $F(1,48) = 5.82$, $p = 0.02$, partial $\eta^2 = 0.11$. Specifically, higher memory performance was observed for stimuli before loss feedback (M = 0.21, SD = 0.85) as compared to stimuli before gain feedback (M = 0.18, SD = 0.81); $t(49) = 3.55$, $p < 0.001$. The same effect was not observed for memory performance on trials after loss feedback (M = 0.19, SD = 0.89) compared with trials after gain feedback (M = 0.18, SD = 0.07); $t(49) = 0.97$, $p = 0.34$ (Fig. 1). There was no significant main effect of age, $F(1,48) = 0.37$, $p = 0.55$ (see Supplementary Results 1 & Supplementary Data 1 for details). Reaction times (RTs), related to emotional salience as well as task-related salience, were analysed to gain further insight into behavioural adaptations to salient events. Regarding RTs to (1) emotional salience there was a significant interaction between the before vs after trials with loss vs. gain feedback, $F(1,48) = 19.40$, $p < 0.001$, partial $\eta^2 = 0.29$: Specifically, RTs slowed down after gain feedback, but sped up after loss feedback (loss feedback: before trials (M = 1.10, SD = 0.03), after trials (M = 1.07, SD = 0.02), gain feedback: before trials (M = 1.04, SD = 0.02), after trials (M = 1.06, SD = 0.02)) (see Supplementary Fig. 5). Given that, in our decorrelated design, loss and gain feedback were comparably

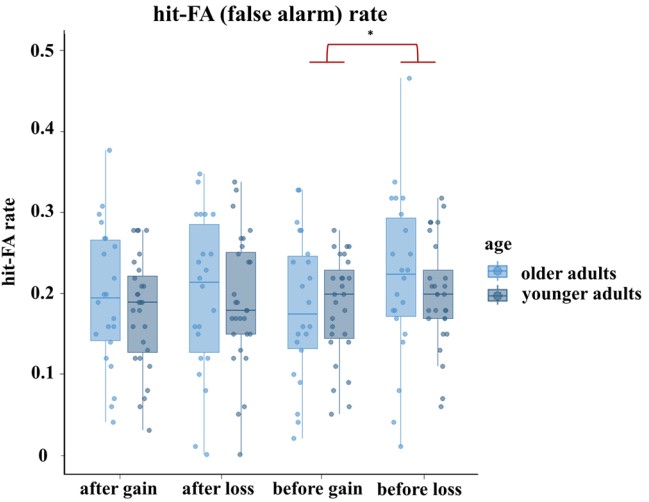

**Fig. 1 | Hit-FA (false alarm) rate.** Hit-FA (false alarm) rate on trials before loss vs. gain feedback and on trials after loss vs. gain feedback for older (brighter blue) and younger (darker blue) adults. The asterisk highlights the statistically significant difference between higher memory performance for stimuli before loss feedback (M = 0.21, SD = 0.85) as compared to stimuli before gain feedback (M = 0.18, SD = 0.81) across age groups, $t(49) = 3.55$, $p < 0.001$ (reproduced from ref. 17, Fig. 2a). Boxplots contain the median (horizontal black line), with the lower and upper parts of the boxes indicating the 25th and 75th percentiles of the underlying hit-FA rate, respectively.

informative for response correctness, this interaction effect might suggest a behaviourally invigorating effect of emotionally salient events in line with animal work showing that LC activity is linked to effortful responding[33]. Effects of (2) task-related saliency on RTs on trials with reversals of the reinforced stimulus category (40 trials in total) were assessed by averaging RTs on three trials before and after reversals, respectively. RTs after reversals were slower (M = 1.05, SD = 0.02) as compared to RTs before the reversal (M = 1.03, SD = 0.03), $F(1,48) = 5.57$, $p = 0.02$, partial $\eta^2 = 0.10$, in both younger as well as older adults, indicating more controlled response behaviour on the first trials of a new reinforced stimulus category. For a complete overview of RT effects related to (1) emotional salience and (2) task-related salience, please see Supplementary Results 1 & Supplementary Data 1.

**fMRI results**

Analyses on fMRI data were conducted to assess the LC and substantia nigra / ventral tegmental area (SN/VTA) response to (1) emotional salience: loss > gain feedback, (2) task-related salience: reversal > no reversal feedback, (3) memory performance: remembered > not remembered and (4) emotional memory performance: remembered before loss feedback > not remembered before loss feedback in younger and older adults, as well as age differences (N = 50) therein (see Table 1). Similarly, cortical, and subcortical regions were examined in the respective contrasts (1-4). Activations in the

# Higher LC activation in older adults

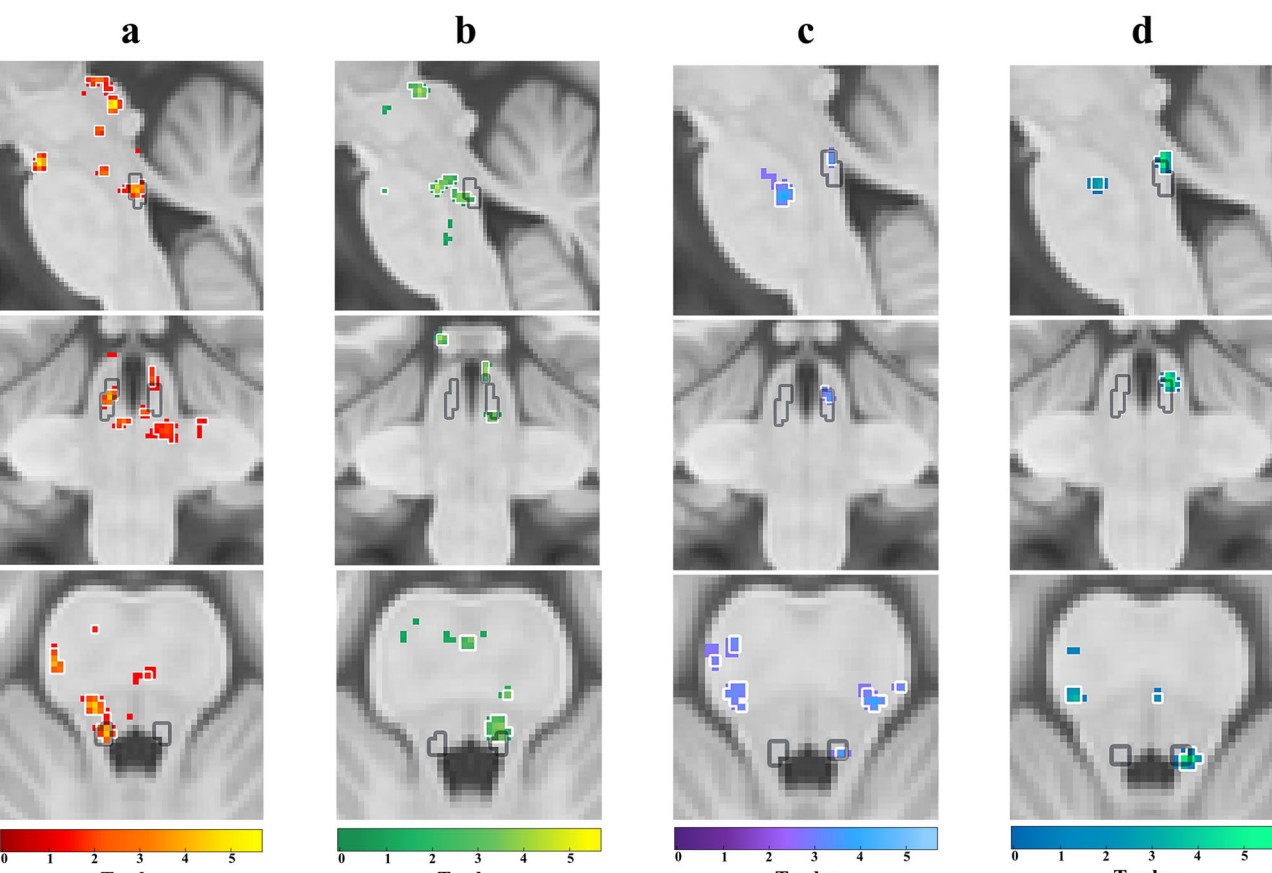

**Fig. 2 | Higher locus coeruleus (LC) activation in older adults.** Higher locus coeruleus (LC) activation in older adults for **a** (1) emotional salience: loss > gain feedback (red-yellow), **b** (2) task-related salience: reversal > no reversal feedback (green-yellow), **c** (3) memory performance: remembered > not remembered (purple-blue) and **d** (4) emotional memory performance: remembered before loss feedback > not remembered before loss feedback (blue-green). Significant activations (**a–d**) shown in each colour with a threshold of $p < 0.005$ (threshold of $p < 0.003$ outlined in white) are in sagittal (first row), coronal (middle row), and axial (bottom row) views, within the LC meta mask (grey) created by ref. [102].

**Fig. 3 | Higher substantia nigra pars reticulata (SNr) activation in older adults.** Higher substantia nigra pars reticulata (SNr) activation in older adults for (2) task-related salience: reversal > no reversal feedback (green-yellow). Activations shown in colour with a threshold of $p < 0.005$ (threshold of $p < 0.003$ outlined in white) are in sagittal (first), coronal (middle), and axial (right) views, within the 'SNrSNcVTA mask' (see Supplementary Fig. 1: *SNr*: dark blue; *SNc*: middle blue; *VTA*: brighter blue; *red nucleus*: red). The black asterisk indicates the significant activation within SNr.

# Higher SNr activation in older adults

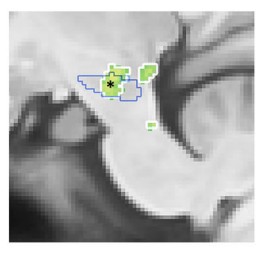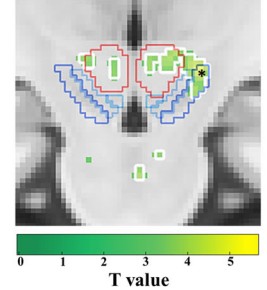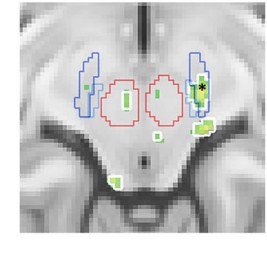

brainstem and midbrain were investigated using an inclusive brainstem mask (see section 'Anatomical masks for second-level analyses').

## Higher LC and SNr activation in older adults during encoding of salient and negative stimuli

In line with our hypothesis, stronger activations in noradrenergic structures were observed during the processing of salient events, and additionally also in GABAergic (SNr) structures. Unexpectedly, this effect was generally more pronounced in older adults (Figs. 2–4). While younger adults did not show significant activations in the LC, older adults showed a

higher activation of the left LC [T = 4.11, pFWE = 0.04 (voxel cut-off $p < 0.005$); pFWE = 0.02 (voxel cut-off $p < 0.003$) (Fig. 2a)] during (1) loss > gain feedback (see Supplementary Table 1). Additionally, older adults also showed higher activation of the right LC and right SNr [*LC*: T = 3.37, pFWE = 0.08 (voxel cut-off $p < 0.005$); pFWE = 0.05 (voxel cut-off $p < 0.003$); *SNr*: T = 4.77, pFWE = 0.02 (voxel cut-off $p < 0.005$); pFWE = 0.02 (voxel cut-off $p < 0.003$) (Figs. 2b and 3)] during (2) reversal > no reversal feedback (see Supplementary Table 4) categories. This dovetails findings from electrophysiological recordings in monkeys showing that the LC responds to relevant or unexpected task events[34]. Age group

# Higher LC activation in older > younger adults

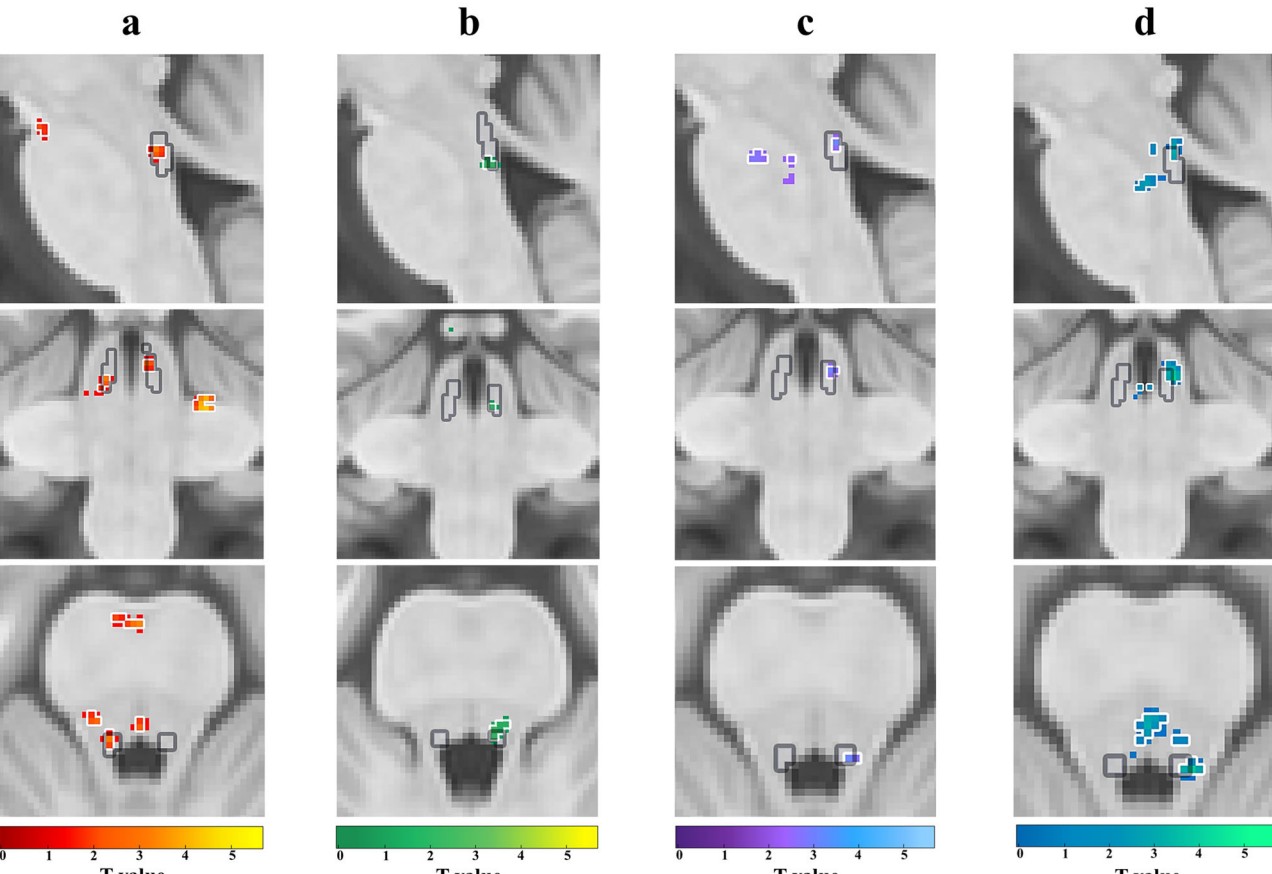

**Fig. 4 | Higher locus coeruleus (LC) activation in older > younger adults.** Higher locus coeruleus (LC) activation in older > younger adults for **a (1)** emotional salience: loss > gain feedback (red-yellow), **b (2)** task-related salience: reversal > no reversal feedback (green-yellow), **c (3)** memory performance: remembered > not remembered (purple-blue) and **d (4)** emotional memory performance: remembered before loss feedback > not remembered before loss feedback (blue-green). Significant activations (**a**–**d**) shown in each colour with a threshold of $p < 0.005$ (threshold of $p < 0.003$ outlined in white) are in sagittal (first row), coronal (middle row), and axial (bottom row) views, within the LC meta mask (grey)[102].

comparisons confirmed the stronger engagement of LC in older adults, showing more engagement of the bilateral LC [left LC: T = 3.4, pFWE = 0.06 (voxel cut-off $p < 0.005$); pFWE = 0.04 (voxel cut-off p < 0.003); right LC: T = 3.14, pFWE = 0.08 (voxel cut-off $p < 0.005$); pFWE = 0.05 (voxel cut-off $p < 0.003$) (Fig. 4a)] during (1) loss > gain feedback (see Supplementary Table 2) in older as compared to younger adults and a trend for higher right LC activation [T = 3.02, pFWE = 0.07 (voxel cut-off $p < 0.005$); pFWE = 0.06 (voxel cut-off $p < 0.003$) (Fig. 4b)] during (2) reversal > no reversal feedback (see Supplementary Table 5) for older adults. No significant emotional and task-related activations in the brainstem were observed in younger adults (see Supplementary Table 3 & 6).

For (2) task-related salience, no correlation was found between the behavioural performance indicators and (a) LC activation but for (b) MTG activation (see Supplementary Fig. 8, Supplementary Results 2; r (16) = 0.62, $p = 0.009$).

## Higher LC activation in older adults for remembering negative stimuli

Older adults additionally exhibited increased LC engagement during later remembered stimuli, particularly if those were associated with negative feedback. Specifically, older adults showed higher activation of the right LC during encoding of later (3) remembered > not remembered stimuli (see Supplementary Table 7) [T = 3.64, pFWE = 0.07 (voxel cut-off $p < 0.005$); pFWE = 0.05 (voxel cut-off $p < 0.003$) (Fig. 2c)] as well as a trend for higher

right LC activation during (4) later remembered stimuli followed by loss as compared to not remembered stimuli followed by loss feedback (see Supplementary Table 11) [T = 4.38, pFWE = 0.07 (voxel cut-off $p < 0.005$); pFWE = 0.06 (voxel cut-off $p < 0.003$) (Fig. 2d)]. Age group comparisons (see Supplementary Table 8 & 12) confirmed greater engagement of the right LC for (3) remembered > not remembered stimuli (see Supplementary Table 8) [T = 3.42, pFWE = 0.08 ($p < 0.005$); pFWE = 0.05 ($p < 0.003$) (Fig. 4c)], as well as for (4) later remembered stimuli followed by loss as compared to not remembered stimuli followed by loss feedback (see Supplementary Table 12) [T = 3.46, pFWE = 0.09 ($p < 0.005$); pFWE = 0.05 ($p < 0.003$) (Fig. 4d)] for older adults as compared to younger adults. No significant memory-related activations in the brainstem were observed in younger adults (see Supplementary Table 9-10 & 13-14). Neither for (1) emotional salience, (3) memory nor (4) emotional memory performance a correlation between the behavioural performance indicators and LC activations was observed (see Supplementary Fig. 6, 9-10).

## Activation in cortical areas

The focus of our study was to examine the brainstem and adjacent areas at higher resolution, which allowed us to investigate only cortical and subcortical activations in limited regions, including parts of the temporal and parietal lobes, amygdala, and HPC, due to the smaller field of view (FoV) (see Fig. 5). Specifically, during (1) loss > gain feedback (see Supplementary Fig. 2; Supplementary Table 15-17), younger and older adults showed

**Fig. 5 | MNI image with an applied partial volume mask.** MNI image with an applied partial volume mask (light gray) in **a** axial, **b** sagittal, **c** coronal view.

## MNI image with an applied partial volume mask

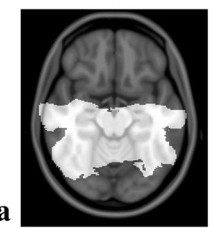
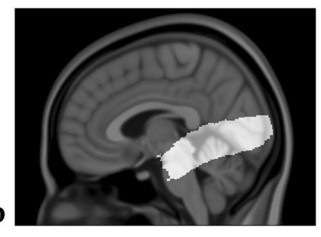
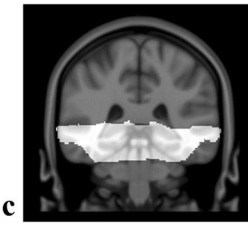

**Fig. 6 | Higher MTG and HPC activation in older > younger adults.** Higher **a, b** middle temporal gyrus (MTG: threshold of $p < 0.005$) and **c** hippocampus activation (HPC: threshold of $p < 0.05$) in older > younger adults for **(1)** emotional salience: loss > gain feedback (red-yellow). HPC activations shown within 'hippocampus-amygdala mask' (see Supplementary Fig. 1: *amygdalae*: rose; *hippocampi*: middle rose; *parahippocampi*: dark rose). Turquoise circles highlight the corresponding significant activations.

## Higher MTG & HPC activation in older > younger adults

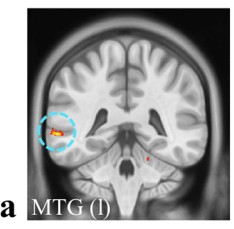
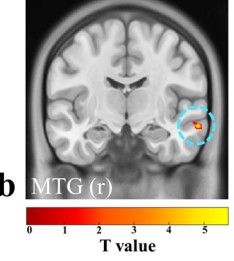
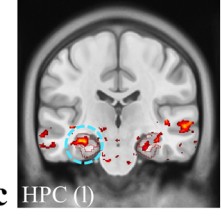

stronger functional activation in the right MTG (younger adults: T = 5.32, pFDR < 0.001; older adults: T = 7.74, pFDR < 0.001). Only older adults showed stronger activation for the left MTG (T = 4.79, pFDR < 0.001). In line with the age group differences in brainstem activations, age group comparisons showed greater engagement of the bilateral MTG (right MTG: T = 4.29, pFDR < 0.04; left MTG: T = 4.54, pFDR < 0.004) for older adults. Similarly, older adults showed a stronger engagement of bilateral MTG (right MTG: T = 8.14, pFDR < 0.001; left MTG: T = 5.70, pFDR = 0.001) during (2) reversal > no reversal feedback (see Supplementary Fig. 3; Supplementary Table 18-20) category while younger adults also showed higher activation in left MTG (T = 5.28, pFDR = 0.02), as well as bilateral STG (right STG: T = 6.61, pFDR < 0.001; left STG: T = 5.54, pFDR < 0.001). No age group differences in MTG or STG activations during reversal > no reversal feedback were observed. In addition to MTG and STG activations, areas supporting visual processing were more activated during salient events in both age groups, including the calcarine cortex (CAL), fusiform gyrus (FuG) and lingual gyrus (LiG) (see Supplementary Results 2, Supplementary Table 15-16, 18-21). Finally, areas known to support memory consolidation and memory-related stimulus processing were preferentially engaged during (2) reversal > no reversal feedback (see Supplementary Fig. 3; Supplementary Table 18-20): Younger adults showed higher activation in left entorhinal cortex (EC) (T = 4.58, pFDR = 0.05) and older adults showed higher activation in the right precuneus (PCUN) (T = 6.59, pFDR < 0.001). Age group comparison showed more engagement of the left PCUN (Older > Younger adults: T = 4.44, pFDR < 0.001) for older adults.

### Hippocampus (HPC)

As salience and memory related LC activations have been known to modulate HPC[29], as well as, amygdala function[35], exploratory analyses in these regions using small volume corrections in anatomical masks were added. During (1) loss > gain feedback (Fig. 6; Supplementary 16), older adults showed stronger functional activations in the left HPC (T = 3.94, pFWE = 0.008) as compared to younger adults. This is in line with the observed stronger LC activation in older adults during loss compared to gain feedback and the better memory for stimuli before loss as compared to gain feedback in both age groups.

### Discussion

The present study investigated age differences in functional activations of brainstem neuromodulatory nuclei during (1) emotional salience, (2) task-

related salience, (3) memory performance and (4) emotional memory performance (Table 1). While no age-related differences in memory emerged, both age groups demonstrated better memory performance for stimuli associated with loss feedback, consistent with previous evidence of better memory for negative events[8,16,36,37]. The unexpected absence of age-related differences in memory performance could be due to floor effects restricting the detection of interindividual differences in our challenging memory task which included greyscale stimuli in the recognition tests which might have made detecting old stimuli more challenging. Indeed, Hit-false alarm rates were comparatively low and high false alarm rates in particular in early recognition tests might indicate weaker memory representations and higher interference (see Hämmerer et al.[17]). Moreover, it is additionally possible that a stronger top-down focus on salient events in older adults (accompanied by stronger top-down regulation of LC activity[38,39]) could compensate for weaker memory representations during encoding and thereby contribute to the absence of age differences in memory performance. Given our necessarily restricted field of view in MRI acquisitions which did not include prefrontal cortices, we are unable to confirm this hypothesis in our imaging data, by e.g. examining age differences in frontal or parietal activations. As a potential support for the hypothesis, we did observe that concurrently acquired pupillometric data showed larger pupil diameters for loss and reversal stimuli in older as compared to younger adults, which might reflect a stronger attentional focus on salient events in older adults (see Hämmerer at al.[17]). However, future imaging studies which systematically manipulate the attentional focus while encoding salient events are needed to confirm these considerations. Contrary to previous evidence of age-related structural decline in neuromodulatory structures, older adults showed increased activations in these brainstem structures compared to younger adults (Table 1). Larger functional activations in older adults are a recurring finding in aging research and typically indicate age-related differences in the emphasis of cognitive processes during fMRI studies, which are often also related to compensating for declining cognitive capacity[40–42]. At the same time, studies show that in older adults, the functional connectivity of the LC with frontoparietal networks decreases, resulting in older adults' attention becoming less selective[43,44]. Additionally, although older adults generally appear to have a bias for recalling positive stimuli[45,46], there is evidence that older adults process negative stimuli in greater detail[47,48] and are more likely to recall them when the stimuli in question provide performance

feedback[49]. The reversal reinforcement learning task's stronger LC engagement in older adults in our study could therefore result from a) compensatory overactivation of the LC or a structure targeting the LC in older adults. The aging brain activates attentional networks, cortical areas, and new networks to compensate functionally for deficits in other areas[42]. Hence, increased LC and HPC activation in older adults might suggest such compensatory efforts. Although memory performance differences between age groups were absent, heightened activations in older adults could imply greater compensatory engagement. However, within older adults, memory performance related to (1) emotional or (2) task-related salience was not correlated to interindividual differences in LC activation (see Supplementary Fig. 7-8). Only higher MTG activation in older adults during (2) task-related salience was associated with better delayed memory performance (see Supplementary Fig. 8), indicating MTG's potential compensatory involvement in older adults' memory processes. Larger samples might be required to link behavioural correlates with functional activations in small brain structures such as the LC. Alternatively, b) stronger functional LC responses could be related to cell loss in the LC, as there is some evidence suggesting that remaining neurons in a shrinking LC might increase firing rates in mice[50]. The lack of correlation between LC integrity and its higher activation in older adults across conditions (see Supplementary Fig. 7-10) suggests more extensive studies for solid validation. Finally, it is conceivable that c) age group differences in task engagement could contribute to differences in functional recruitment between age groups. The simple reinforcement learning task with deterministic feedback and few reversals (40 trials) may have caused younger adults to exert less effort than older adults, consistent with observations of youth-like activations in high-performing older adults, particularly during easy tasks[51]. Unfortunately, our limited FoV prevented us from comparing task-focused parietal or frontal activations across age groups to further investigate the extent of youth-like activations. Additionally, the LC also plays an essential role in the maintenance of vascular functions. The individual variability of the arteries supplying the LC (specific patterns of LC vascularization such as dilation and constriction) could consequently affect cognition, potentially explaining differences in the age-groups, but this currently requires further research[52,53]. In terms of the brainstem or midbrain's functional responses during (1) emotional salience, the LC was more activated in processing loss rather than gain feedback, and feedback indicating a reversal in task conditions as compared to feedback indicating no reversal (Figures 2–4). Enhanced LC responses to emotionally salient events in our study align with animal studies showing its preferential firing to negative events such as foot shocks[16,33] and in vivo findings linking LC integrity and memory for negative events in older adults[17,32,37]. Further converging evidence for the LC-NE systems focus on emotional or salient events is provided by measures of pupil diameter, which may indicate underlying phasic LC activity[9,17] but are not exclusively associated with phasic LC responses (see Joshi et al.[54]). Larger pupil diameters are typically observed for emotionally negative stimuli such as negative feedback[17,55–58]. Indeed, higher pupil dilations to negative feedback have also been observed in this study, in both, younger as well as older adults (Hämmerer et al.[17]). Although negative feedback was less frequent (about 33% of trials) in our task, the LC's stronger responses to negative feedback were not due differences in informativeness, because our non-probabilistic reversal reinforcement learning task balanced losing and gaining trials on reversal feedback. To the extent that pupil dilations are indicative of underlying LC activity, this finding is extended by a recent pupillometry study that demonstrated higher pupil dilations during negative performance feedback in younger and older adults while controlling for informativeness as well as frequency of feedback[57]. This would suggest that the mere negative value of an event is sufficient to elicit LC activation. Given that we observed faster responses after loss but not gain feedback in both younger and older adults, higher LC activations during loss feedback could be behaviourally related to a general response invigoration. This interpretation needs to be supported by further in vivo imaging work but is consistent with animal studies showing that phasic LC responses are particularly related to

effortful response execution[59]. Regarding functional responses in the brainstem or midbrain to (2) task-related salience, increased activation of both the LC and SNr were observed in older adults during feedback indicating task reversals (cf. Figures 2–4). As dopaminergic nuclei (VTA, SNc) and GABAergic nucleus (SNr)[60] as well as LC are reciprocally connected[16], a concurrent activation is both anatomically and functionally plausible. Simultaneous electrical recordings in mice's ventral VTA and LC showed that both regions were more active in a novel environment, with the LC showing a more rapid decrease in response over time compared with the VTA[61]. Similarly, the SN has been observed to show phasic activation in response to unexpected and novel events, suggesting coordinated responses by noradrenergic and dopaminergic structures[62,63]. Specifically, animal work has shown that (a) SNr neurons (in rodents) and (b) LC neurons (in monkeys) exhibit increased firing rates in response to task-relevant cues such as (a) Go and Stop cues[64] or (b) infrequent deviant stimuli in so-called oddball tasks that required specific behavioural responses[9]. Thus, higher LC and SNr activation during reversals is consistent with animal work showing that both structures are activated in response to novel, unexpected, or deviant stimuli that might require response adaptation. Consistent with the need to adjust stimulus category preferences after reversals in our task, both younger and older adults responded more slowly on trials after reversals. Moreover, in older adults, midbrain and brainstem nuclei were more involved not only in processing response feedback but also in subsequent memory effects during stimulus presentation. Specifically, they showed higher LC activations for (3) memory performance, especially for (4) emotional memory performance, during encoding compared to younger adults. The results confirm the involvement of the LC in encoding and remembering of emotional events, as the LC is known to support the long-term memory formation, notably for negative events[29,30,65,66] via NE release in LC target areas like amygdala and HPC, which are important for encoding and retrieving emotional events[26,29,67]. In line with this, previous studies demonstrated that older adults with higher rostral LC integrity exhibited better memory performance[18], consistent with the finding that rostral LC contrast in the elderly is associated with the thickness of widespread cortical regions[68] and loss of rostral LC is associated with poorer memory performance[66]. It is interesting to speculate whether the observed activation patterns of LC, namely emotional salience preferentially engaging larger portions of the left LC and memory related processes engaging the right LC (Fig. 2), might relate to differences in projection patterns of the LC. A study in older people showed a loss of rostral-like connectivity of the LC and differences in the spatial properties of the LC gradient associated with poorer emotional memory, with left rostral-like connectivity reduced compared to right connectivity in people with higher levels of anxiety and depression[66]. In animals, caudal portions of the LC preferentially project to the spinal cord and cerebellum, while rostral portions tend to project to the cerebral cortex and forebrain, including amygdala and hippocampus[69–72]. If these projection patterns can be transferred to humans, our observation of right more rostral LC activations for memory-related processes would be consistent with these different projection patterns. Furthermore, both younger and older adults exhibited greater engagement of the right MTG during (1) emotional salience, with older adults showing stronger activation in bilateral MTG. During (2) task-related salience older adults also showed a stronger bilateral activation of MTG, while younger adults showed higher activation in left MTG. Given the involvement of the MTG in semantic cognition and processing of negative words[73–75] the higher activation in elders in the MTG on (1) emotional and (2) task-related salience may reflect similar responses to salient stimuli in our task. While the STG is primarily associated with phonological processing (for a review, see Bhaya-Grossman and Chang[76]), it has also been shown to respond to rare stimuli in oddball tasks, suggesting attention-related processes[77,78]. Thus, the observed bilateral activation of the STG in younger adults during (2) task-related salience may reflect its involvement in task-related salience and attentional demands. Finally, older adults showed higher activation in the right precuneus (PCUN) and compared to younger adults, they also

showed higher activation in the left PCUN during (2) task-related salience. This may reflect greater processing and increased formation of new memories during task-related saliency in elders, as PCUN plays a crucial role during encoding[79]. While the underlying causes of hyperactivity in the PCUN are yet to be determined, it could serve as an early functional marker of AD pathological changes, despite its lack of correlation with AD bio-markers in cerebrospinal fluid (CSF)[80]. Additionally, younger adults showed higher EC activation during reversals (an indicator of cognitive flexibility), which appears to be correspondingly stronger in younger individuals[81] and may be consistent with facilitated transfer of encoded information to the HPC[82]. The higher hippocampal activation in older compared to younger adults for (1) emotional salience may therefore indicate a stronger engagement of areas supporting memory encoding during salient events[82,83]. As a necessary limitation given our emphasis on a high-resolution data acquisition, results observed in this study reflect activations in a partial brain volume (Fig. 5). The joint area available for the second level analyses could therefore not include other brain regions of interest such as the insula, anterior cingulate cortex (ACC) or thalamus which are known to be part of the salience network and important target areas for the LC[84]. Similarly, compensatory relationships in supporting areas like parietal or frontal cortices which input the LC could not be investigated[85,86]. Future studies using for instance stronger magnetic field strengths should remedy this. Since sex differences in the LC-NE system and memory performance are known[87,88] (for review see ref. 89) we additionally investigated potential sex differences in LC activation for (1) emotional salience, (2) task-related salience, (3) memory performance, and (4) emotional memory performance. The analysis approach did not yield any significant clusters in the brainstem when investigating sex differences in the given contrasts. Consequently, we are unable to provide any findings regarding sex differences. Likewise, we did not find any potential sex differences in behavioural memory performance (see Supplementary Results 2). Finally, given the increasingly better understood age differences in neurovascular function, particularly in relation to altered vasoreactivity and blood oxygen consumption, age group differences in the neurovascular coupling underlying the BOLD response should be expected[90]. While the various contributions to altered BOLD responses in ageing as well as interindividual differences therein are currently not yet completely determined[90–92], baseline tasks which should engage more similar processes in different age groups (e.g., finger tapping) revealed generally lower SNRs in the BOLD response of older as compared to younger adults[93]. It is thus conceivable that age differences, pointing towards larger effects in older adults, in our study might underestimate existing age differences somewhat. Taken together, our study demonstrates the feasibility of investigating age differences in functional LC involvement during a cognitive task known to rely on noradrenergic function. In line with animal work on LC function, we observed greater activation of the LC during task-related and emotionally significant events and during memory encoding of the latter. Interestingly, a stronger LC engagement in response to emotionally and task-related salient events was associated with differential reaction time effects: acceleration and deceleration, respectively. This result highlights that the LC, along with other salience-indicating structures, might operate within a network of brain structures aiming to regulate context-specific adaptive responses. In contrast to accumulating findings showing a structural decline in the LC in ageing, LC engagement was generally stronger in older adults in our study. Given the comparable behavioural performance between younger and older adults, this might indicate a potential compensatory overactivation of the LC in older adults. Future studies should build on these results and investigate the interplay between salience and compensatory processes independently, e.g., by using tasks manipulating attentional focus and salience simultaneously. Moreover, it is essential to investigate whether changes in functional responses of the LC are a typical sign of healthy ageing or can serve as a biomarker for mild cognitive impairment (MCI) and AD[19,31]. Therefore, further studies should additionally assess CSF or blood-based biomarkers as pre-symptomatic indicators of dementia-related brain pathology[20,94], solidifying the link between age-related and pathological declines as well as functional assessments of brainstem and midbrain nuclei.

## Methods
### Dataset
The data reported in this article are part of a study that included structural and functional brainstem imaging, pupillometric recording, and an emotional memory task (see Hämmerer et al., 2018). The present article focuses on the functional brain imaging data during the emotional memory task; for a previous report on the structural brainstem data and their relationship to memory task performance see Hämmerer et al.[17]. A total of 50 English speaking people, 28 healthy younger adults (16 females) with a mean age of 23.14 (range: 20 to 31 yrs., SD = 3.18) and 22 healthy older adults (12 females) with a mean age of 67.68 (range: 65 to 84 yrs., SD = 5.68) participated in the study (for sample description see Supplementary Table 22). Suitability for the study was assessed using a telephone questionnaire administered by research assistants during recruitment and again in person by radiographers before the experimental examination. Specifically, subjects who were unsuitable for scanning (e.g. metallic implants, claustrophobia) and subjects with a history of neurological (e.g. neurodegenerative diseases) or psychiatric disorders were excluded. Subjects were right-handed (Old-field questionnaire lateralization quotient >80)[95]. The study was approved by the local ethics committee (University College London ethics reference no. 5506/001) and written informed consent was obtained from each subject prior to participation. All ethical regulations relevant to human research participants were followed. Subjects received a payment of £50 for their participation, including a bonus payment of £6 based on task performance (all subjects performed at a high level and received the bonus payment). An abbreviated version of the Raven's Progressive Matrices[96] was used to examine whether subjects matched known markers of age differences in adult fluid intelligence[97]. Performance was assessed as the number of correctly solved matrices of the 18 given matrices within 20 min. Due to changes in the test design, only 19 younger adults completed the fluid intelligence tasks. The younger adults performed better than older adults [t(39) = 3.45, $p < 0.001$], indicating that subjects were consistent with the known age differences in fluid intelligence. Subjects performed a reversal reinforcement learning task (cf. Supplementary Fig. 12) to assess the impact of salient events on memory while undergoing fMRI recording (57–61 min). Specifically, subjects learned through positive (smiling face, two-point gain) and negative (sad face, two-point loss) feedback on both forced and free choice trials whether indoor or outdoor scenes were rewarded or not, as well as whether a reversal in the rewarded scene category has occurred (forced trials: 66.26 loss trials and 60.64 gain trials on average; free choice trials: 13.08 loss trials and 120.04 gain trials on average; 303 trials in total; 6 runs × 9.62 min). Importantly, using forced choice trials our task design allowed for examining two different types of saliences in processing choice feedback (loss versus gain and reversal versus no reversal) by balancing loss and gain feedback in particular on reversal trials. The reversal reinforcement learning task was followed by an unannounced immediate (60 min after encoding) and delayed (4–6 h after encoding) memory test to assess whether memory for scene stimuli before loss vs. gain feedback and reversal vs. no reversal points differed. Further details about the stimulus material, task design as well as study procedure can be found in Hämmerer et al.[17].

### s/fMRI data acquisition
MRI data were acquired on a 3 T Tim Trio System (Siemens Healthineers, Erlangen, Germany) with a standard 32-channel radiofrequency (RF) coil. Structural and functional imaging sequences were optimised for LC imaging. 3D multi-echo FLASH structural images were acquired as part of a modified multiparameter mapping protocol[98]. Anisotropic voxel sizes (0.4 × 0.4 × 3 mm³) aiming to match the stick-like shape of the LC were acquired in a slab oriented parallel to the back of the brainstem aiming to have the longer voxel dimension coincide with that of the LC[4]. In addition, a whole-brain isotropic T1-weighted FLASH image (voxel sizes: 0.75 iso-tropic, FOV 240 × 240 × 64 3 mm³) was acquired as an anatomical scan for

image registration. Further details on the structural MRI data acquisition can be found in Hämmerer et al.[17]. Resolution and coverage of the fMRI data were optimized both to measure the small LC (which is only about 1-3 mm wide and ~ 15 mm long[22]) with sufficient resolution and to have a field of view (FoV) that allows assessment of the HPC and amygdala along with other brainstem nuclei (Fig. 6). For this purpose, a 6 cm wide angulated 3D T2*-weighted EPI (TE = 37.3 ms, TR = 76 ms, FA 15° water-selective excitation, parallel imaging with GRAPPA factor 2 in the phase-encoded EPI direction, Bandwidth 1395 Hz/Px, FOV 192 mm × 192 mm × 48 mm, with a 1.5 mm isotropic voxel size, 32 partitions plus 25% oversampling) was positioned as described above. The volume acquisition time was 3.04 s. During the reversal reinforcement learning task each subject had a total of 1140 measurements spaced across 6 runs of 190 measurements each, resulting in 6 runs of about 9.62 min per subject (first 5 measurements of each run were discarded). The full sample also comprises 5 pilot subjects with a slightly different run separation of a total of 1200 measurements, spaced across 5 runs of 240 measurements each, resulting in 5 runs of about 12.16 min.

### LC segmentation

The left and right LC were each manually segmented by two raters in the anatomical MRI images using ITK-Snap[86]. LC integrity was assessed as signal intensity within segmentations averaged across left and right LC, normalised with respect to a nearby area in the brainstem. Note that due to poor LC visibility in two subjects (older [right LC] and younger [left LC]), segmentations for only one side of the LC were included (see Hämmerer et al.[17] including Supplementary Fig. 4 for more details on LC segmentations and contrast analyses).

### fMRI data pre-processing and dedicated post-processing pipeline for spatially precise brainstem imaging

fMRI data pre-processing and statistical modelling was done using Statistical Parametric Modelling 12 (SPM12; Wellcome Centre for Human Neuroimaging, University College, London, UK, 2012) as well as Advanced Normalization Tools (ANTs) v.2.1.0 software package (http://stnava.github.io/ANTs/). The raw DICOM data were converted to NIfTI images, while preserving the original image parameters. The pre-processing of the functional data was performed in SPM12 and included realignment, unwarping, and smoothing (2 mm FWHM). Without registering or normalising pre-processed data, first level contrasts were calculated in native space (see below for more details). Also, all whole-brain T1w images were used to generate a study-specific template, using the antsMultivariateTemplateConstruction2.sh function in ANTs with default parameters except the rigid-body registration option on. Registration and normalisation of functional and structural data to MNI space (ICBM 152 nonlinear asymmetric template T1w, 1 mm resolution [mni_icbm152_t1_tal_nlin_asym_09c.nii][99]) followed a pipeline developed for assuring sufficient spatial transformation precision in the brainstem area (see Yi et al.[24], Fig. 2 for an overview). Specifically, after correcting for B0 field inhomogeneity of the partial-volume brain T1w images (N4BiasFieldCorrection from ANTs[100]), the following steps were carried out: To match the above-mentioned MNI space resolution, neuromelanin-sensitive structural images (FLASH) were re-sampled to 1-mm isotropic voxel size using the mri_convert function in FreeSurfer (Version 7.1; http://surfer.nmr.mgh.harvard.edu/, Martinos Center for Biomedical Imaging, Charlestown, Massachusetts; see Yi et al.[24]; Fig. 2f). Using antsRegistrationSyN.sh, the whole-brain T1w images in the native space were non-linearly registered to the study specific template before being non-linearly registered to the MNI space. Concatenated transformation matrices and deformation fields from these steps, using antsApplyTransforms.sh, the whole-brain T1w images were transformed onto the MNI space. Afterwards, the whole-brain T1w images were rigidly registered to the individual mean EPI images (see Yi et al.[24]; Fig. 2d). Additionally, the structural T1w slab and manually drawn individual LC segmentation in the space of the T1w slab was rigidly registered to the partial volume brain T1w images (using antsRegistrationSyN.sh). To align the individual LC segmentation to the whole-brain T1w images rigidly, the same

transformation matrix from this registration step was applied to the LC mask. Finally, combinations of the above-described transformations were applied to the mean EPI images and the first-level statistical contrast images as well as the LC masks in each of their respective native space in a single step and were transformed to the MNI space (see Yi et al.[24]; Fig. 5–1, 5–2) non-linearly (using antsApplyTransforms.sh). Therefore, group level analyses in MNI space were possible while assuring high precision of spatial transformations and reducing bias due to multiple interpolations. All structural and mean EPI images were transformed using the fourth-order B-Spline interpolation, while the statistical contrast data were transformed using linear interpolation, and individual LC segmentations were transformed by using nearest neighbour interpolation.

### Quality checks for assuring sufficient spatial transformation precision of structural and functional LC imaging data

To evaluate the quality of spatial transformation of structural and functional LC imaging data across subjects, guidelines following Yi et al.[24] were used. For assessing the precision of functional LC imaging data, 8 different landmarks were placed by two independent raters on individual mean functional images in MNI space in the brainstem area (see Yi et al.[24] for more details) (Fig. 7). To ensure a similar approach to setting the landmarks, raters were first trained together on an independent training dataset. Afterwards, to ensure independent ratings, raters worked separately on the present dataset, while balancing across the two raters which part of the data was rated first to account for possible training effects. Both raters had experience with rating several different datasets. Sørensen–Dice coefficient (DSC) score was calculated to assess the consistency across the two raters (0 indicates no spatial overlap, while 1 indicates a complete overlap). The following DSC scores resulted for the 8 landmarks: nucleus ruber (l) = 0.70, nucleus ruber (r) = 0.70, periaqueductal grey = 0.62, perifastigial sulcus = 0.53, outline brainstem (l) = 0.34, outline brainstem (r) = 0.33, 4th ventricle border (l) = 0.72, 4th ventricle border (r) = 0.64, representing a good overlap of the two raters (for age-related differences in 8 brainstem landmarks' mean functional images in MNI space see Supplementary Results 3). Note that overlap in landmarks for the brainstem outline is generally lower as more degrees of freedom exist in the anterior-posterior direction[24]. As can be seen in Fig. 7, quality checks suggest a good spatial precision in transforming functional LC data into MNI space, with deviations as assessed in landmarks not exceeding 2.5 mm in the LC (blue bar graphs in Fig. 7), which is the assumed average width of the LC based on post-mortem data[22]. Segmentations delineating the LC in multi-parameter mapping scans for structural LC imaging were performed by two independent raters in native space (see Hämmerer et al.[17] for more details), where the DSC score was 0.72, indicating the overlap between the two raters in identifying voxels belonging to the LC (see Hämmerer et al.[17]). An overlay of the binary LC segmentations after transformation to MNI space is shown as a heatmap in Fig. 8. In addition, to assess the precision of the alignment of LC segmentations in MNI space across subjects, distances across subjects for the left and right LC centroid voxels were calculated for each slice of the LC segmentation (Fig. 7). Deviations assessed across subjects and averaged across slices within subjects did not exceed 2 mm overall, the median slice-wise distance on the left side was 0.80 mm (Mdn ± MAD = 0.80 ± 0.16) (younger adults: 0.80 mm [Mdn ± MAD = 0.80 ± 0.14], older adults: 0.73 mm [Mdn ± MAD = 0.73 ± 0.19]), and on the right side 0.82 mm (Mdn ± MAD = 0.82 ± 0.21) (younger adults: 0.89 mm [Mdn ± MAD = 0.88 ± 0.26], older adults: 0.76 mm [Mdn ± MAD = 0.76 ± 0.18]). Deviations did not differ between left and right side (F(1,95) = 0.005, p = 0.94) and only showed a trend for being larger in younger adults (F(1,95) = 3.4, p = 0.07). Note that deviations in LC positions between subjects likely do not solely stem from imprecisions in spatial transformations, as LC positions in native space also differ between subjects by on average about 1.45 mm (left LC) and 0.96 mm (right LC) (see Yi et al.[24]; Fig. 5), as evident in post-mortem and structural LC imaging data[22]. Spatial deviations across subjects after transformation thus likely

# Quality checks of structural and functional LC imaging data

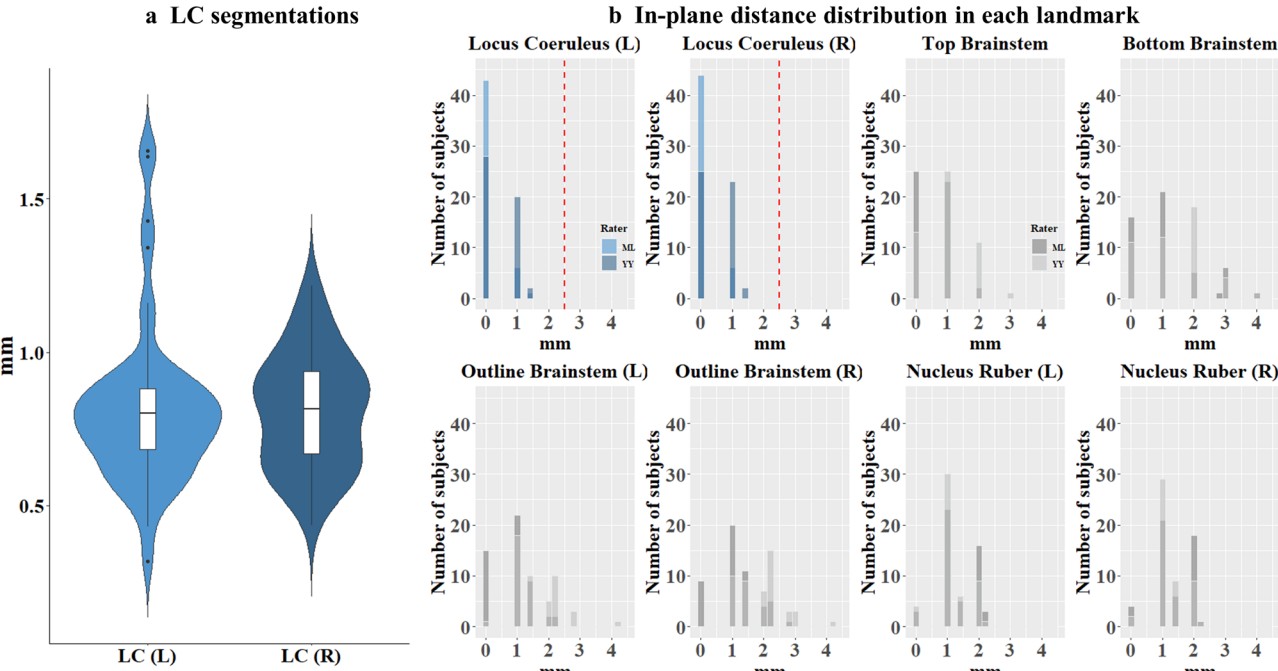

**a** LC segmentations

**b** In-plane distance distribution in each landmark

**Fig. 7 | Quality checks of structural and functional LC imaging data. a** The distribution of inter-subject distances for the left and right LC centroid voxels of the aggregated LC meta mask[102] and the MNI-transformed LC segmentations for individual subjects are shown in violin plots. Boxplots within the violin plot show error bars with 95% confidence interval. In-plane distance is calculated separately for the left and right LC, slice by slice, and averaged across slices to obtain a value per subject and left or right LC segmentation (right: $M\pm$SD=0.81 ± 0.19, IQR = 0.28; left: $M \pm$ SD=0.81 ± 0.27, IQR = 0.21). **b** Histograms of in-plane distances between single-subject landmarks and landmarks defined on the MNI template. The dashed red line indicates the typical width of the LC (2.5 mm[22] below which deviations should fall[24], in fact median deviations all fell below 1 mm.

**Fig. 8 | Heatmap of transformed individual LC segmentations.** Heatmap of transformed individual LC segmentations in the group space (from left to right: axial, sagittal, coronal view). The blue line indicates the LC meta mask created by ref. 102. The maximum overlap across segmentations within the LC meta mask is at 62% and the minimum overlap at 1.6%.

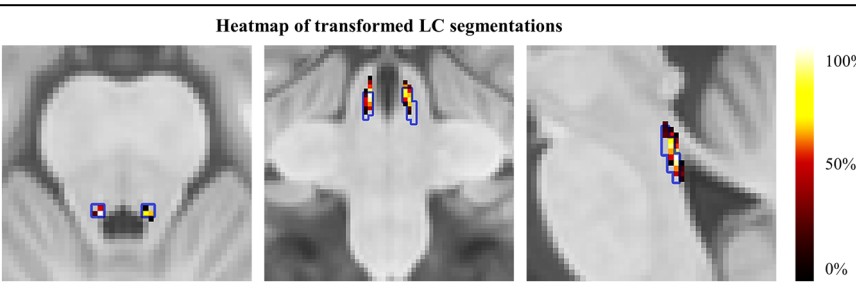

Heatmap of transformed LC segmentations

represent a mixture of biological variations in LC position and imprecision in spatial transformations.

## Anatomical masks for second-level analyses

Anatomical masks of study-relevant brain regions (see Supplementary Fig. 1) were used in region of interest (ROI) - specific analyses for precise delineations of functional activation patterns with small-volume correction (SVC). For nuclei in the brainstem and midbrain, substantia nigra pars reticulata (SNr, label 9), substantia nigra pars compacta (SNc, label 7), ventral tegmental area (VTA, label 11) and red nucleus (label 8) masks were extracted from a high-resolution probabilistic in vivo atlas by Pauli et al.[101] (https://identifiers.org/neurovault.collection:3145). Anatomical templates were already in the MNI template space adopted in our study[99] and only needed to be binarized for use of ROI-specific analyses (using mni_binarize). The binarized masks were thresholded at 0.20 to combine the different templates into a mask that matched the anatomical definition of the SN substructures ('SNredVTA mask' - from now on, this nomenclature will refer to these structures: SNr, SNc, VTA, red nucleus) (using SPM image calculator). For the LC, the LC meta mask (https://osf.io/sf2ky/) by Dahl

et al.[102] was used, which is a combination of several already existing individual LC masks[14,103–106] and also consistent with LC dimensions reported in post-mortem studies[22] (for more details see Dahl et al.[102]). The LC meta mask[102] was non-linearly co-registered to the MNI template space adopted in our study (using antsRegistration) with nearest neighbour interpolation (using antsApplyTransforms). As can be seen in Fig. 8, the LC meta mask[102] shows good agreement with LC segmentations in our study. In addition, as the study also focuses on salience and memory-related functional activations, a combined bilateral mask including the hippocampus, parahippocampus and amygdala referred to as 'hippocampus-amygdala mask' was created based on the Cerebrum Atlas (CerebrA) by Manera et al.[107], since it provides non-linear registration of Mindboogle atlas[108] to high resolution MNI-ICBM 2009c[99] space of cortical and subcortical regions (see Manera et al.[107]). Besides left (label 99) and right (label 48) hippocampus as well as left (label 70) and right (label 19) amygdala, left (label 69) and right (label 18) para hippocampal regions were extracted (using fslmaths). Individual templates were binarized (using mri_binarize) and combined (using SPM image calculator) to create a final bilateral mask. Anatomical templates were already in the MNI template space adopted in our study[99]

and only needed to be binarized for use of ROI - specific analyses (using mni_binarize). Finally, a 'grey matter mask' based on the MNI template adopted in our study[99] and a 'brainstem mask' based on CerebrA[107] were used as an implicit mask in second-level analyses. The grey matter mask was created by segmenting the MNI template[99] (using SPM segment, Bias FWHM 30 mm cut-off). For creating this brainstem mask, left (label 62) and right (label 11) brainstem as well as left (label 77) and right (label 26) ventral diencephalon were extracted from the CerebrA[107] (using fslmaths), were binarized (using mri_binarize) and combined (using SPM image calculator) to one mask representing brainstem and midbrain regions.

## Behavioural analyses

Using repeated measures ANOVA and paired-samples t-tests across both age groups, analyses of behavioural data were conducted to compare memory performance and RTs of both age groups for single and double scene stimuli that occurred on trials before and after loss vs. gain feedback, and on trials before and after a reversal. Memory was measured as the mean of the hit-FA (false alarms) rate across both recognition tests. These analyses were carried out using SPSS version 28.0.0.1 (IBM; https://www.ibm.com/analytics/de/de/technology/spss)[109] and MATLAB version R2020b (The MathWorks)[110]. Correlation analyses between significant LC, MTG activations, LC integrity and memory performance in the elderly were carried out in RStudio version 2022.02.3 using cor() function for Spearman´s Rank correlation, corr.test() (package psych[111]) to adjust with Bonferroni correction for multiple comparisons and corrplot() (package corrplot[112]) for visualisation. Figures were created using the R package ggplot2[113].

## fMRI data first-level and second-level analyses

As the focus of the study was to investigate the processing of salient events in a reinforcement learning task, the main contrasts of interest were (1) loss feedback > gain feedback as an indicator of emotional salience and (2) reversal feedback > no-reversal feedback as an indicator of task-related salience. Furthermore, to investigate memory effects, the contrasts (3) remembered stimuli > not remembered stimuli as in indicator of memory performance and (4) remembered stimuli before loss feedback > not remembered stimuli before loss feedback as an indicator of emotional memory performance were investigated. To address these questions, four event-related General Linear Models (GLMs) were implemented which allowed investigation of these contrasts in younger adults, older adults and age group differences in these contrasts. Specifically, GLM 1 assessing emotional salience included loss and gain feedback timepoints while controlling for reversal feedback timepoints. GLM 2 assessing effects of task-related salience included regressors for reversal and no reversal feedback while controlling for timepoints of loss feedback. GLM 3 assessing memory performance included timepoints of remembered and not remembered stimuli during stimulus presentation, and GLM 4 assessing emotional memory performance included regressors of remembered or not remembered stimuli before gain or loss feedback during stimulus presentations. To account for irrelevant task-related effects, GLMs included regressors indicating where stimuli or feedback (depending on the GLM) were part of a free or forced choice trial (one or two stimuli to choose from), as well as the onset of the fixation cross between stimulus and feedback presentations and left and right response time points. For an overview of all regressors included in the respective GLMS, see Table 1, while the time course of the effect size is shown in Supplementary Fig. 11. For an overview of the main fMRI results, see Supplementary Table 23. All sets of GLMs also contained regressors of no interest (6 regressors for movement, 14 regressors for physiological data like breathing and pulse). Finally, because high resolution functional images are more susceptible to movement artefacts during recording, individual volumes with movement exceeding a pre-set threshold were excluded from the statistical analyses by modelling them with an individual volume regressor in the first level GLM. Criteria for a volume exclusion were a displacement exceeding .75 mm (half a voxel) or 0.5° in rotation (Lawson et al.[114]). Movement artefacts during recording did not differ between healthy younger and older healthy subjects as assessed by mean distance and

degree in displacement; $t(47) = 1.17$, $p = 0.25$ (two healthy older adults did not have volumes exceeding exclusion criteria). The first 5 (dummy) volumes were not included in the GLM analyses. First level contrasts effects were then included in second level analyses which assessed contrasts of interest within as well as between age groups using one sample t-tests two sample t-tests, respectively. Given the small size of our target structures in the brainstem and midbrain, significant activations were assessed using anatomical masks of the LC and a combined mask of SNc, SNr, VTA and red nucleus for small volume corrections. Activations in cortical and subcortical areas were examined using an inclusive grey matter mask. For the confirmatory small volume corrected analyses, significance assessments were corrected for multiple comparisons using family wise error correction (FWE), which is a more conservative measure assessing the ratio of falsely rejected tests to all tests performed. For the more exploratory assessments of significant clusters in cortical and subcortical areas, multiple comparisons were corrected using the false discovery rate correction (FDRc), which is based on the ratio of falsely rejected tests to all rejected tests and more sensitive in detecting clusters of activation[115–120]. Given the comparatively smaller voxel sizes (1.5 mm isotropic) and the relatively lower SNR per voxel, less conservative voxel cut-offs of $p < 0.005$ were used for cortical and subcortical areas to increase sensitivity. Furthermore, for target structures in the brainstem and midbrain, a more conservative voxel cut-off of $p < 0.003$ in addition to $p < 0.005$ was used to indicate the contribution of more reliably activated areas in the brainstem and midbrain (see white lines in Figs. 2–3,5). The analysis procedure described above partially resulted in no suprathreshold clusters for brainstem, midbrain, cortical, and subcortical areas (see Supplementary Results 2 for details).Formularbeginn

## Data availability

Data (Supplementary Data 1) supporting this study are available on OSF at https://osf.io/h5pgr/. All other data are available on reasoned request from the corresponding author.

## Code availability

The codes supporting this study are available on OSF at https://osf.io/h5pgr/.

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

## Funding

## Competing interests
The authors declare no competing interests.
