## [Peer Review File · Communications Biology]

Reviewers' comments:

Reviewer #1 (Remarks to the Author):

This paper reports fMRI findings of a sophisticated memory task that investigates age-related and emotion/salience-related differences in locus coeruleus (LC) activation, along with the substantia nigra and related cortical regions. The authors used a specifically developed method for looking a predefined brain regions and contrasted between different feedback conditions and found differences, particularly in the brain stem, between young and old subjects.

Major comments:

1. While these findings are interesting, the results would be more compelling if the authors demonstrated raw activations of the LC (i.e. activation time and percent BOLD signal change) to individual tasks. Supplemental figure 11 shows t-values for individual subjects but not activations. Interesting, a number of subjects show negative t-values.
2. The differences in activation between the old and young subjects could also be the result of differences in hemodynamic response. This factor should be considered in interpreting the results.
3. There is clear laterality in some of the activation differences in the LC. I wonder if that is a result of low sensitivity or some neurobiological underpinning.

Reviewer #2 (Remarks to the Author):

I previously reviewed this manuscript by Ludwig and colleagues and it has been resubmitted with significant revisions. Ludwig and colleagues used rigorous imaging methodology to investigate age-related differences in functional LC activation as it relates to salience (emotional and task-related) and memory (neutral and emotional) in older and younger adults. The authors carefully addressed comments from all reviewers in the previous round and have improved the manuscript significantly, though I have additional comments:

1. As previously brought up, the lack of observed difference in memory performance between older and younger adults is surprising and is not sufficiently addressed. The authors briefly referenced it in the Discussion p. 13 ("although memory performance differences between age groups were absent"), but I believe this warrants additional discussion. The authors may offer reasons why they think this is the case in the text and whether conclusions should be considered carefully given that there was age difference.
2. When asked to address sex differences, the authors say there were insufficient numbers of subjects (57% of 28 YA were female and 55% of 22 OA were female) to address sex differences. What is insufficient regarding these numbers? Based on a power analysis?
3. Figures 2-5 should include a color bar indicating what the color maps correspond to. P-value? They do this in Figure 8 but not in the other figures.
4. I understand that the format of the journal has Methods at the end, but I find the Results difficult to follow without greater context of the various conditions. I imagine the reader will have difficulty understanding what terms such as "trials before vs. trials after feedback" (what kind of feedback?) and "loss vs gain" in the context of the study without some explanation before. A quick description of these terms/conditions in the introduction would help with the flow and overall readability. Perhaps something similar to table 1 could be presented to show the reader these different conditions.

5. There is no table breaking down the basic demographics of the subjects. It may help with readability to have a table that shows the n's of each age group, sex breakdown, any other collected demographics, and perhaps the key findings summarized in that table (similar to table 1, but showing directionality of what they found).

6. The Methods are very detailed and rigorous for MRI processing and statistical analyses but lack more information about the subjects included themselves. Participants are referred to as "healthy" but that is the only characterization of them. For older adults in particular, was there any cognitive screening? Medical/physical exclusions? Education level? I understand many of these details are available in other papers, but this paper should stand alone and include better characterization of the sample.

Reviewer #3 (Remarks to the Author):

In the revised manuscript, the sparse changes made following earlier reviewer comments mean my major concerns (clarity of writing and statistical analysis) remain.

For example, a major concern I requested to be addressed was the clarity of the analysis ('LC activation'). I recognise the references, but these must be unpacked in methods and briefly discussed in the results. FWE and FDRc are techniques with nuances that are unclear in the current format. This should be clear, as BOLD activation forms the basis of all results.

As my concerns were not with the manuscript's 'impact', changing the journal does not address the matter. I am of the opinion that a serious overhaul of the text is required. Nevertheless, as my background is in a different field, I defer my judgement to the other reviewers if they believe the manuscript is in a state that provides clear value to the community.

Minor:

Fig.1: Please state state 'reproduced' from Hämmerer et al., 2018, Fig. 2a

Supp. Fig. 11: Showing individual means is a great start. Nevertheless, here is another example of unclear analysis: what is the stated F-test calculated? Is it between individual means of the two groups? Are you calculating the F-test between the first 28 values in row 1 and the final 22 values? Also, colouring by an individual is not informative; please colour by t-value (if colouring at all).

Functional *LOCUS COERULEUS IMAGING* to investigate an ageing noradrenergic system

Mareike Ludwig^{abc}, Yeo-Jin Yi^{ac}, Falk Lüsebrink^{cde}, Martina F. Callaghan^f, Matthew J. Betts^{abc}, Renat Yakupov^c, Nikolaus Weiskopf^{fgh}, Raymond J. Dolan^{fi}, Emrah Düzel^{acj*}, Dorothea Hämmerer^{abcfjk*}

^aInstitute of Cognitive Neurology and Dementia Research, Otto-von-Guericke University Magdeburg, Magdeburg, Germany; ^bCBBS Center for Behavioral Brain Sciences, Magdeburg, Germany; ^cGerman Center for Neurodegenerative Diseases (DZNE), Magdeburg, Germany; ^dBiomedical Magnetic Resonance, Faculty of Natural Sciences, Otto-von-Guericke University, Magdeburg, Germany; ^eNMR Methods Development Group, Max Planck Institute for Human Cognitive and Brain Sciences, Leipzig, Germany; ^fWellcome Centre for Human Neuroimaging, UCL Queen Square, Institute of Neurology, University College London, 12 Queen Square, London WC1N 3AR, UK; ^gDepartment of Neurophysics, Max Planck Institute for Human Cognitive and Brain Sciences, Stephanstraße 1a, 04103 Leipzig, Germany; ^hFelix Bloch Institute for Solid State Physics, Faculty of Physics and Earth Sciences, Leipzig University, Linnéstraße 5, 04103 Leipzig, Germany; ⁱMax Planck University College London Centre for Computational Psychiatry and Ageing Research, London WC1B 5EH, United Kingdom; ^jInstitute of Cognitive Neuroscience, University College London, London WC1N 3AZ, United Kingdom; ^kDepartment of Psychology, University of Innsbruck; *shared last authorship

Response letter for reviewers

Dear Reviewers,

We sincerely appreciate the valuable time you have taken to thoroughly read and carefully review the manuscript we submitted. We greatly appreciate your thoughtful insights and constructive feedback, as they play an invaluable role in improving the quality and rigor of our research. We have addressed all comments and included respective changes in the manuscript.

All the changes have been indicated in **green** in this document as well as in the manuscript and Supplementary Material.

Responses to reviewers:

- Reviewer 1: page 2-5
- Reviewer 2: page 6-13
- Reviewer 3: page 14-15

Reviewer #1 (Remarks to the Author)

This paper reports fMRI findings of a sophisticated memory task that investigates age-related and emotion/salience-related differences in locus coeruleus (LC) activation, along with the substantia nigra and related cortical regions. The authors used a specifically developed method for looking a predefined brain regions and contrasted between different feedback conditions and found differences, particularly in the brain stem, between young and old subjects.

Major comments:

1. While these findings are interesting, the results would be more compelling if the authors demonstrated raw activations of the LC (i.e. activation time and percent BOLD signal change) to individual tasks. Supplemental figure 11 shows t-values for individual subjects but not activations. Interesting, a number of subjects show negative t-values.

Thank you for your feedback. Regarding the previous Supplementary Figure 11, it is indeed interesting that a number of subjects showed negative t-values which might possibly suggest that gains, or non-reversals might have carried more salience for specific individuals. However, at present there is no established reliability for interpreting individual-specific functional LC activations, which likely requires longer acquisitions per individual in particular for the small brainstem nuclei (Gordon et al., 2017). In the context of the present dataset, we are thus focusing on analyses on the group level, which either indicated a larger LC activation for negative or reversal timepoints or a non-significant difference from zero. It will be very desirable to in the future acquire highly sampled data of individuals to investigate these possibly individual-specific differences in salience processing in the LC.

Regarding the suggestion to provide time courses of raw activations of the LC for the respective tasks, we now provide time courses of the **effect size** for a) **(1)** emotional salience: loss > gain feedback, b) **(2)** task-related salience: reversal > no reversal feedback, c) **(3)** memory performance: remembered > not remembered and d) **(4)** emotional memory performance: remembered before loss feedback > not remembered before loss feedback. The reason for providing effect size time courses within a trial rather than percent signal change in raw data within a trial is that the reported task-related results of functional LC activations show effects based on GLMs which control for concurrent experimental effects (that is e.g. investigating loss > gain while controlling for reversals). We now include the time courses for the task-related effects (as e.g., in Boorman et al. 2009) indicated below in the supplementary materials (Supplementary Figure 11).

It should be noted that the time courses of the effect sizes in Supplementary Figure 11 should be interpreted with caution due to the small number of voxels activated in the locus coeruleus (LC) (cf. Supplementary Table 2,5,8,12) which might render temporal extrapolations of functional activations more prone to noise. Time courses of the effect sizes of LC activations are shown based on voxel cut-offs of $p < 0.005$ as in the original GLMs. Beta values per contrast are shown for older adults (dotted black line), younger adults (solid black line) as well as contrast estimates of the effect older adults > younger adults (solid red line); shaded areas indicate ± 1 SE. Yellow lines indicate the onset of relevant task events, ensuing time courses in particular in a 5 sec time window should be indicative of effects captured in the BOLD response in the original GLMs.

Time course of the effect size

Supplementary Figure 11. Time course of the effect size for a) (1) emotional salience: loss > gain feedback, b) (2) task-related salience: reversal > no reversal feedback, c) (3) memory performance: remembered > not remembered and d) (4) emotional memory performance: remembered before loss feedback > not remembered before loss feedback for Locus Coeruleus (LC) activations (voxel cut-offs of $p < 0.005$). Beta values per contrast (a-d) for older adults (dotted black line) and younger adults (solid black line) and (a-d) the coefficient estimates for older adults > younger adults (solid red line) are plotted; shaded areas +/- 1 SE. Yellow lines indicate the onset of relevant task events.

2. The differences in activation between the old and young subjects could also be the result of differences in hemodynamic response. This factor should be considered in interpreting the results.

We thank the reviewer for raising this important point. We now included a paragraph in the discussion which addresses potential effects of age differences in vascular function on the BOLD response with relevance for our study.

We have therefore added in the **Discussion**:

“[...] Finally, given the increasingly better understood age differences in neurovascular function, particularly in relation to altered vasoreactivity and blood oxygen consumption, age group differences in the neurovascular coupling underlying the BOLD response should be expected⁹⁵. While the various contributions to altered BOLD responses in ageing as well as interindividual differences therein are currently not yet completely determined⁹⁵⁻⁹⁷, baseline tasks which should engage more similar processes in different age groups (e.g., finger tapping) revealed generally lower SNRs in the BOLD response of older as compared to younger adults⁹⁸. It is thus conceivable that age differences, pointing towards larger effects in older adults, in our study might underestimate existing age differences somewhat. [...]“

3. There is clear laterality in some of the activation differences in the LC. I wonder if that is a result of low sensitivity or some neurobiological underpinning.

Thank you very much for this interesting comment. It is indeed interesting that, in older adults, emotional saliency appears to preferentially engage the left LC while memory related processes engage the right LC.

Postmortem data based on neuromelanin-pigmented LC cells would not suggest that the left LC in humans is systematically larger than the right LC (Baker et al., 1989; German et al., 1988; ChanPalay and Asan, 1989; Ohm et al., 1997). Note that laterality differences can however be observed in structural imaging studies of the LC, which typically find larger structural volumes for the left LC on Siemens scanners and for the right LC on Philips scanners, possibly related to the scanner-type specific B0-field directions which are opposite (Tona et al., 2017). Similarly, structural imaging studies showed that the left LC contrast was greater than the right LC contrast; but there were no age differences (see Figure S3; Bachmann et al., 2021). This is consistent with the findings of Liu et al. (2019), who acquired data on a Siemens scanner and reported a higher maximum contrast for the left LC compared to the right LC and also showed that older adults had a greater variance in overall LC contrast compared to younger adults (Liu et al., 2019). However, at present we are not aware of a study comparing these scanner-type effects systematically in the same sample and it is unclear how these would translate into functional results. Taken together we don't have postmortem evidence to believe that the left or the right LC should be differentially affected in ageing, although it is of course conceivable that subparts of the LC are functionally specialized independent of age-related structural changes. It is as of yet unknown whether left and right LC project differentially to emotionally relevant (e.g. amygdala) and memory-relevant (e.g. hippocampus) brain structures in humans. We do know however, at least from animal studies, that the rostral LC is preferentially projecting to the cerebral cortex, including amygdala and hippocampus, whereas caudal LC portions tend to project to the spinal cord and the cerebellum (Loughlin et al., 1986; Schwarz and Luo, 2015; Ungerstedt, 1971; Samuels and Szabadi 2008b). In addition, a study in older people

showed a loss of rostral-like connectivity of the LC and differences in the spatial properties of the LC gradient that were associated with poorer emotional memory (Veréb et al., 2023). Likewise, they showed that left rostral-like connectivity was reduced compared to right connectivity in people with higher levels of anxiety and depression (Vereb et al., 2023). In this regard, our observation that activations related to memory encoding preferentially occur in the right parts of the LC would be consistent with the animal and human literature on right rostral LC projections relevant for supporting memory encoding.

We have added these considerations to the **Discussion**:

“[...] loss of rostral LC is associated with poorer memory performance⁷⁷. It is interesting to speculate whether the observed activation patterns of LC, namely emotional salience preferentially engaging larger portions of the left LC and memory related processes engaging the right LC (cf. Figure 2), might relate to differences in projection patterns of the LC. A study in older people showed a loss of rostral-like connectivity of the LC and differences in the spatial properties of the LC gradient associated with poorer emotional memory, with left rostral-like connectivity reduced compared to right connectivity in people with higher levels of anxiety and depression⁷⁷. In animals, caudal portions of the LC preferentially project to the spinal cord and cerebellum, while rostral portions tend to project to the cerebral cortex and forebrain, including amygdala and hippocampus⁸⁰⁻⁸³. If these projection patterns can be transferred to humans, our observation of right more rostral LC activations for memory-related processes would be consistent with these different projection patterns.

Reviewer #2 (Remarks to the Author)

I previously reviewed this manuscript by Ludwig and colleagues and it has been resubmitted with significant revisions. Ludwig and colleagues used rigorous imaging methodology to investigate age-related differences in functional LC activation as it relates to salience (emotional and task-related) and memory (neutral and emotional) in older and younger adults. The authors carefully addressed comments from all reviewers in the previous round and have improved the manuscript significantly, though I have additional comments:

1. As previously brought up, the lack of observed difference in memory performance between older and younger adults is surprising and is not sufficiently addressed. The authors briefly referenced it in the Discussion p. 13 (“although memory performance differences between age groups were absent”), but I believe this warrants additional discussion. The authors may offer reasons why they think this is the case in the text and whether conclusions should be considered carefully given that there was age difference.

We apologize for not having addressed this sufficiently in the first revision. Indeed, it is important to discuss the lack of the observed differences in memory performance between older and younger adults.

We have therefore expanded the **Discussion** as follows:

“[...] The unexpected absence of age-related differences in memory performance could be due to floor effects restricting the detection of interindividual differences in our challenging memory task which included greyscale stimuli in the recognition tests which might have made detecting old stimuli more challenging. Indeed, Hit-false alarm rates were comparatively low and high false alarm rates in particular in early recognition tests might indicate weaker memory representations and higher interference (see Hämmerer et al.¹⁷). Moreover, it is additionally possible that a stronger top-down focus on salient events in older adults (accompanied by stronger top-down regulation of LC activity^{50,51}) could compensate for weaker memory representations during encoding and thereby contribute to the absence of age differences in memory performance. Given our necessarily restricted field of view in MRI acquisitions which did not include prefrontal cortices, we are unable to confirm this hypothesis in our imaging data, by e.g. examining age differences in frontal or parietal activations. As a potential support for the hypothesis, we did observe that concurrently acquired pupillometric data showed larger pupil diameters for loss and reversal stimuli in older as compared to younger adults, which might reflect a stronger attentional focus on salient events in older adults (see Hämmerer et al.¹⁷). However, future imaging studies which systematically manipulate the attentional focus while encoding salient events are needed to confirm these considerations. [...]”

2. When asked to address sex differences, the authors say there were insufficient numbers of subjects (57% of 28 YA were female and 55% of 22 OA were female) to address sex differences. What is insufficient regarding these numbers? Based on a power analysis?

Thank you for the question! We would like to take this opportunity to correct our statement. Given the overall sample of 50 subjects and the numbers of subject per age group, it is reasonable to also look at the biological sex differences. We have now analyzed potential biological sex differences.

We have now added these results to the **Supplementary Results 2, Sex differences:**

Regarding potential sex differences in **LC activation** we additionally investigated the main contrasts of interests: **(1)** loss feedback > gain feedback as an indicator of **emotional salience** and **(2)** reversal feedback > no-reversal feedback as an indicator of **task-related salience**, **(3)** remembered stimuli > not remembered stimuli as an indicator of **memory performance** and **(4)** remembered stimuli before loss feedback > not remembered stimuli before loss feedback as an indicator of **emotional memory performance**. Activations in the brainstem were investigated using an inclusive brainstem mask (see section *Anatomical masks for second-level analyses*). Given the small size of our target structures in the brainstem activations were assessed using anatomical masks of the LC. This described analysis procedure did not result in any suprathreshold clusters for brainstem in the contrasts listed above when examining sex differences, so unfortunately, we cannot report any results on sex differences.

Regarding potential sex differences in **behavioural memory performance**, we have additionally run a repeated measures ANOVA for stimuli that occurred on trials before and after loss vs. gain feedback. There was no significant main effect of sex, $F(1,48) = 0.05$, $p = 0.83$. Likewise, when controlling for age and sex, there was no significant effect of sex, $F(1,46) = 0.05$, $p = 0.83$ and also no significant interaction between sex and age, $F(1,46) = 0.37$, $p = 0.55$.

We furthermore included the following section in **Discussion:**

“[...] Future studies using for instance stronger magnetic field strengths should remedy this. Since sex differences in the LC-NE system and memory performance are known^{92,93} (for review see⁹⁴) we additionally investigated potential sex differences in LC activation for (1) emotional salience, (2) task-related salience, (3) memory performance, and (4) emotional memory performance. The analysis approach did not yield any significant clusters in the brainstem when investigating sex differences in the given contrasts. Consequently, we are unable to provide any findings regarding sex differences. Likewise, we did not find any potential sex differences in behavioural memory performance (see Supplementary Results 2).”

3. Figures 2-5 should include a color bar indicating what the color maps correspond to. P-value? They do this in Figure 8 but not in the other figures.

Thank you for your valuable feedback. We have added the color bars representing the T-Values in Figure 2-5 accordingly.

Higher LC activation in older adults

Figure 2. Higher locus coeruleus (LC) activation in older adults for **a) (1)** emotional salience: loss > gain feedback (red-yellow), **b) (2)** task-related salience: reversal > no reversal feedback (green-yellow), **c) (3)** memory performance: remembered > not remembered (purple-blue) and **d) (4)** emotional memory performance: remembered before loss feedback > not remembered before loss feedback (blue-green). Significant activations (a-d) shown in each colour with a threshold of $p < 0.005$ (threshold of $p < 0.003$ outlined in white) are in sagittal (first row), coronal (middle row), and axial (bottom row) views, within the LC meta mask (grey) created by Dahl et al.³⁷.

Higher SNr activation in older adults

Figure 3. Higher substantia nigra pars reticulata (SNr) activation in older adults for **a) (2)** task-related salience: reversal > no reversal feedback (green-yellow). Activations shown in colour with a threshold of $p < 0.005$ (threshold of $p < 0.003$ outlined in white) are in sagittal (first), coronal (middle), and axial (right) views, within the ‘SNrSNcVTA mask’ (see Supplementary Fig. 1: SNr: dark blue; SNc: middle blue; VTA: brighter blue; red nucleus: red). The black asterisk indicates the significant activation within SNr.

Higher LC activation in older > younger adults

Figure 4. Higher locus coeruleus (LC) activation in older > younger adults for **a) (1)** emotional salience: loss > gain feedback (red-yellow), **b) (2)** task-related salience: reversal > no reversal feedback (green-yellow), **c) (3)** memory performance: remembered > not remembered (purple-blue) and **d) (4)** emotional memory performance: remembered before loss feedback > not remembered before loss feedback (blue-green). Significant activations (a-d) shown in each colour with a threshold of $p < 0.005$ (threshold of $p < 0.003$ outlined in white) are in sagittal (first row), coronal (middle row), and axial (bottom row) views, within the LC meta mask (grey) Dahl et al.³⁷.

Higher MTG & HPC activation in older > younger adults

Figure 5. Higher (a,b) middle temporal gyrus (MTG: threshold of $p < 0.005$) and (c) hippocampus activation (HPC: threshold of $p < 0.05$) in older > younger adults for (1) emotional salience: loss > gain feedback (red-yellow). HPC activations shown within ‘hippocampus-amygdala mask’ (see Supplementary Fig. 1: *amygdalae*: rose; *hippocampi*: middle rose; *parahippocampi*: dark rose). Turquoise circles highlight the corresponding significant activations.

4. I understand that the format of the journal has Methods at the end, but I find the Results difficult to follow without greater context of the various conditions. I imagine the reader will have difficulty understanding what terms such as “trials before vs. trials after feedback” (what kind of feedback?) and “loss vs gain” in the context of the study without some explanation before. A quick description of these terms/conditions in the introduction would help with the flow and overall readability. Perhaps something similar to table 1 could be presented to show the reader these different conditions.

We thank the reviewer for this comment. We therefore made the following changes in the introduction to ensure a better understanding of the different conditions.

“In our study, we wanted to build upon these findings by assessing whether (1) (emotionally) salient events such as negative feedback (loss feedback > gain feedback) are associated with increased LC activation, (2) whether task-related salient events, such as condition reversals (reversal feedback > no reversal feedback), are associated with increased LC activation, and whether (3-4) LC activation during such events contributes to memory performance (remembered > not remembered) for salient events (remembered before loss feedback > not remembered before loss feedback; see Table 1).”

5. There is no table breaking down the basic demographics of the subjects. It may help with readability to have a table that shows the n’s of each age group, sex breakdown, any other collected demographics, and perhaps the key findings summarized in that table (similar to table 1, but showing directionality of what they found).

Thank you for your feedback. We have now created a table with details about the sample and included it in Supplementary Table 22. We have also added a separate overview of the main fMRI results, based on the format of Table 1, in Supplementary Table 23.

In section **Dataset** we refer to Supplementary Table 22: “[...] participated in the study (for sample description see Supplementary Table 22).”

In section **fMRI data first-level and second-level analyses** we now refer to Supplementary Table 23, which includes an overview of the results: “[...] For an overview of all regressors included in the respective GLMS, see Table 1. For an overview of the main fMRI results, see Supplementary Table 23.”

Sample description

	older adults	younger adults
number of subjects	22	28
age (mean±SD)	67.68 ± 5.68	23.14 ± 3.18
gender (F/M)	12 / 10	16 / 12
Raven’s matrices (mean±SD)	14.09 ± 2.43	16.26 ± 1.37

Supplementary Table 22. Overview of sample description. A shortened version of Raven's matrices was used as a measure of fluid intelligence (older adults (N = 22); younger adults (N = 19). The values indicate the correct answers from a total of 18 matrices.

Types of event-related GLMs	contrast of interest	fMRI results
(1) emotional salience	loss feedback	younger adults
	>	▪ right MTG
	gain feedback	older adults
		▪ left LC (Fig. 2a)
		▪ bilateral MTG
		older > younger
		▪ bilateral LC (Fig. 4a)
		▪ bilateral MTG (Fig. 5a-b)
		▪ left HPC (Fig. 5c)
(2) task-related salience	reversal feedback	younger adults
	>	▪ left MTG
	no reversal feedback	▪ bilateral STG
		▪ left EC
		older adults
		▪ right LC (Fig. 2b)
		▪ right SNr (Fig. 3)
		▪ bilateral MTG
		▪ right PCUN
		older > younger
		▪ right LC (Fig. 4b)
		▪ left PCUN
(3) memory performance	remembered	younger adults
	>	/
	not remembered	older adults
		▪ right LC (Fig. 2c)
		older > younger
		▪ right LC (Fig. 4c)
(4) emotional memory performance	remembered before loss	younger adults
	feedback	/
	>	older adults
	not remembered before loss	▪ right LC (Fig. 2d)
	feedback	older > younger
		▪ right LC (Fig. 4d)

Supplementary Table 23. Four types of event-related GLMs with corresponding contrasts of interest and overview of the main fMRI results of younger, older and the group comparison older > younger adults. Additional fMRI results (e.g., CAL, FuG, LiG) are mentioned in Supplementary Results 2.

6. The Methods are very detailed and rigorous for MRI processing and statistical analyses but lack more information about the subjects included themselves. Participants are referred to as “healthy” but that is the only characterization of them. For older adults in particular, was there any cognitive screening? Medical/physical exclusions? Education level? I understand many of these details are available in other papers, but this paper should stand alone and include better characterization of the sample.

Thank you for this feedback. Indeed, we should have given more details on the sample, accordingly we have now added more details on the sample in section **Dataset**:

“[...] participated in the study (for sample description see Supplementary Table 22). Suitability for the study was assessed using a telephone questionnaire administered by research assistants during recruitment and again in person by radiographers before the experimental examination. Specifically, subjects who were unsuitable for scanning (e.g. metallic implants, claustrophobia) and subjects with a history of neurological (e.g. neurodegenerative diseases) or psychiatric disorders were excluded. Subjects were right-handed (Oldfield questionnaire lateralization quotient >80)⁹⁰. An abbreviated version of the Raven's Progressive Matrices⁹¹ was used to examine whether subjects matched known markers of age differences in adult fluid intelligence⁹². Performance was assessed as the number of correctly solved matrices of the 18 given matrices within 20 minutes. Due to changes in the test design, only 19 younger adults completed the fluid intelligence tasks. The younger adults performed better than older adults [$t(39) = 3.45, p < 0.001$], indicating that subjects were consistent with the known age differences in general cognitive abilities.

Subjects performed a reversal reinforcement learning task[...]

Finally, following a previous suggestion of the reviewer, we have now also softened the claim of establishing a biomarker in the title as this would indeed require extensive reliability and predictive tests. The new title now reads:

Functional *LOCUS COERULEUS IMAGING* to investigate an ageing noradrenergic system

Reviewer #3 (Remarks to the Author)

In the revised manuscript, the sparse changes made following earlier reviewer comments mean my major concerns (clarity of writing and statistical analysis) remain.

For example, a major concern I requested to be addressed was the clarity of the analysis ('LC activation'). I recognise the references, but these must be unpacked in methods and briefly discussed in the results. FWE and FDRc are techniques with nuances that are unclear in the current format. This should be clear, as BOLD activation forms the basis of all results.

We thank you for your feedback and apologies for not having made this clearer in the previous round of revisions. We have now clarified the differences between FWE and FDRc more clearly in the following **fMRI results** section:

“Given the small size of our target structures in the brainstem and midbrain, significant activations were assessed using anatomical masks of the LC and a combined mask of SNc, SNr, VTA and red nucleus for small **volume corrections. Activations in cortical and subcortical areas were examined using an inclusive grey matter mask. For the confirmatory small volume corrected analyses, significance assessments were corrected for multiple comparisons using family wise error correction (FWE), which is a more conservative measure assessing the ratio of falsely rejected tests to all tests performed. For the more exploratory assessments of significant clusters in cortical and subcortical areas, multiple comparisons were corrected using the false discovery rate correction (FDRc), which is based on the ratio of falsely rejected tests to all rejected tests and more sensitive in detecting clusters of activation³⁴⁻³⁹.**”

As my concerns were not with the manuscript's 'impact', changing the journal does not address the matter. I am of the opinion that a serious overhaul of the text is required. Nevertheless, as my background is in a different field, I defer my judgement to the other reviewers if they believe the manuscript is in a state that provides clear value to the community.

Thank you for your feedback - We have now incorporated more clarifications in the manuscript and structural revisions of the manuscript which we hope will make its content more accessible to readers.

Minor:

Fig.1: Please state state 'reproduced' from Hämmerer et al., 2018, Fig. 2a

Thank you for your feedback. We have adjusted the wording in Figure 1 accordingly:

“[...] (**reproduced** Hämmerer et al., 2018, Fig. 2a [...])”

Supp. Fig. 11: Showing individual means is a great start. Nevertheless, here is another example of unclear analysis: what is the stated F-test calculated? Is it between individual means of the two groups? Are you calculating the F-test between the first 28 values in row 1 and the final 22 values? Also, colouring by an individual is not informative; please colour by t-value (if colouring at all).

Thank you for your feedback and the opportunity to clarify our analysis. The F-test we conducted served to evaluate the equality of variances between the two age groups. We have decided to remove the overview of significant activations (based on t-values), as the raw bold response was desired.

REVIEWERS' COMMENTS:

Reviewer #1 (Remarks to the Author):

The authors have addressed my concerns to the best of their abilities given the data presented. I believe the manuscript cannot be further improved given the data.

Reviewer #2 (Remarks to the Author):

The revision by Ludwig and colleagues was thorough and I appreciate their additional analyses, edits, and thoughtful responses to my review. They have addressed the majority of my initial comments, and I just have a few remaining thoughts below.

1. The focus on neurodegenerative disease (ND) in the abstract ("has promise as an early biomarker for ND") seems a bit of a reach, as none of the subjects included had ND diagnoses (Methods even state that ND was explicitly excluded on p. 21), nor were they sufficiently characterized in terms of their cognition (apart from a fluid intelligence task), functional status, or biomarker profiles (This is nicely listed as a future direction by the authors in the discussion). I suggest they frame the paper in the Abstract in the context of aging or cognitive aging rather than ND.

2. On p.6 "fMRI results" there appears to be a great deal of description of Methods. Some of this may be necessary for readability (given the journal format of Results before Methods), and I appreciate the authors adding some methods up front to improve comprehension as I mentioned in an earlier review. But other parts, such as the description of multiple comparisons, description of exploratory analyses, etc. may best be moved to the Methods section. In fact, in re-reading the "fMRI results" section starting on p. 5 and ending on p.6, I do not find any actual results presented- it is entirely a description of Methods! In my opinion, the readability and take-home messages from the results would be clearer and more streamlined by moving a great deal of this verbiage out of the Results section.

3. Similar to point 2 above, there is also a lot of contextualization of findings in the Results that is best left for the Discussion (e.g., p. 7 "This is in line with animal studies showing.."; p. 7 "This dovetails findings from electrophysiological.."; p.9 "In line with this, animal studies show that NE release.."; other examples throughout). Since the authors chose to separate Results and Discussion sections, it would benefit the reader to streamline the paper and simply present results in the Results section, and provide interpretation/context in the Discussion, unless they feel this is critical for the paper. I ultimately leave this up to the authors and editor, but combined with my point 2 above, I feel the manuscript would benefit from some streamlining/reorganization to increase readability.

Minor

- Top p.4 "cognitively normal younger and older adults" I would avoid using the phrase "cognitively normal" since it implies a more thorough evaluation of cognition/function, whereas these subjects were administered a single test of fluid intelligence. Also, did the authors compare all subjects performance on the test to see if they were within normative expectations? I didn't see this in the text

- Why have the authors chosen to bold certain words or phrases throughout? It seems random and inconsistent (e.g., "left MTG" vs "bilateral MTG"; sometimes brain regions are bolded, sometimes conditions are bolded, sometimes important take home messages are bolded). Minor point but just something to consider.

- Larger font for activation scale and "T-value" in figures

Reviewer #3 (Remarks to the Author):

The authors have addressed my major concerns in their revised manuscript. The temporal GLM regressions and statistics table are a welcome addition.

Functional locus coeruleus imaging to investigate an ageing noradrenergic system

Mareike Ludwig^{abc}, Yeo-Jin Yi^{ac}, Falk Lüsebrink^{cde}, Martina F. Callaghan^f, Matthew J. Betts^{abc}, Renat Yakupov^c, Nikolaus Weiskopf^{fgh}, Raymond J. Dolan^{fi}, Emrah Düzel^{acj*}, Dorothea Hämmerer^{abcfjk*}

^aInstitute of Cognitive Neurology and Dementia Research, Otto-von-Guericke University Magdeburg, Magdeburg, Germany; ^bCBBS Center for Behavioral Brain Sciences, Magdeburg, Germany; ^cGerman Center for Neurodegenerative Diseases (DZNE), Magdeburg, Germany; ^dBiomedical Magnetic Resonance, Faculty of Natural Sciences, Otto-von-Guericke University, Magdeburg, Germany; ^eNMR Methods Development Group, Max Planck Institute for Human Cognitive and Brain Sciences, Leipzig, Germany; ^fWellcome Centre for Human Neuroimaging, UCL Queen Square, Institute of Neurology, University College London, 12 Queen Square, London WC1N 3AR, UK; ^gDepartment of Neurophysics, Max Planck Institute for Human Cognitive and Brain Sciences, Stephanstraße 1a, 04103 Leipzig, Germany; ^hFelix Bloch Institute for Solid State Physics, Faculty of Physics and Earth Sciences, Leipzig University, Linnéstraße 5, 04103 Leipzig, Germany; ⁱMax Planck University College London Centre for Computational Psychiatry and Ageing Research, London WC1B 5EH, United Kingdom; ^jInstitute of Cognitive Neuroscience, University College London, London WC1N 3AZ, United Kingdom; ^kDepartment of Psychology, University of Innsbruck; *shared last authorship

Response letter for reviewers

Dear Reviewer,

We sincerely thank you for the valuable time you have taken again to thoroughly read and carefully review the manuscript we submitted. We have included all suggested changes and made appropriate alterations to the manuscript.

All the changes have been indicated in **yellow** in this document as well as in the manuscript and Supplementary Material.

Reviewer #2 (Remarks to the Author):

The revision by Ludwig and colleagues was thorough and I appreciate their additional analyses, edits, and thoughtful responses to my review. They have addressed the majority of my initial comments, and I just have a few remaining thoughts below.

1. The focus on neurodegenerative disease (ND) in the abstract (“has promise as an early biomarker for ND”) seems a bit of a reach, as none of the subjects included had ND diagnoses (Methods even state that ND was explicitly excluded on p. 21) , nor were they sufficiently characterized in terms of their cognition (apart from a fluid intelligence task), functional status, or biomarker profiles (This is nicely listed as a future direction by the authors in the discussion). I suggest they frame the paper in the Abstract in the context of aging or cognitive aging rather than ND.

We thank the reviewer for this suggestion and decided to reframe the abstract:

“[...] Moreover, older adults exhibited increased LC activation compared to younger adults, indicating possible compensatory overactivation of a structurally declining LC in ageing. **Our study shows that assessment of LC function is a promising biomarker of cognitive aging.**”

2. On p.6 “fMRI results” there appears to be a great deal of description of Methods. Some of this may be necessary for readability (given the journal format of Results before Methods), and I appreciate the authors adding some methods up front to improve comprehension as I mentioned in an earlier review. But other parts, such as the description of multiple comparisons, description of exploratory analyses, etc. may best be moved to the Methods section. In fact, in re-reading the “fMRI results” section starting on p. 5 and ending on p.6, I do not find any actual results presented- it is entirely a description of Methods! In my opinion, the readability and take-home messages from the results would be clearer and more streamlined by moving a great deal of this verbiage out of the Results section.

We thank the reviewer for the careful re-reading and valuable feedback. We agree with this and have moved the following section to methods “fMRI data first-level and second-level analyses”:

“[...] **Given the small size of our target structures in the brainstem and midbrain, significant activations were assessed using anatomical masks of the LC and a combined mask of SNc, SNr, VTA and red nucleus for small volume corrections. Activations in cortical and**

subcortical areas were examined using an inclusive grey matter mask. For the confirmatory small volume corrected analyses, significance assessments were corrected for multiple comparisons using family wise error correction (FWE), which is a more conservative measure assessing the ratio of falsely rejected tests to all tests performed. For the more exploratory assessments of significant clusters in cortical and subcortical areas, multiple comparisons were corrected using the false discovery rate correction (FDRc), which is based on the ratio of falsely rejected tests to all rejected tests and more sensitive in detecting clusters of activation³⁴⁻³⁹. Given the comparatively smaller voxel sizes (1.5 mm isotropic) and the relatively lower SNR per voxel, less conservative voxel cut-offs of $p < 0.005$ were used for cortical and subcortical areas to increase sensitivity. Furthermore, for target structures in the brainstem and midbrain, a more conservative voxel cut-off of $p < 0.003$ in addition to $p < 0.005$ was used to indicate the contribution of more reliably activated areas in the brainstem and midbrain (cf. white lines in Figure 2-3,5). The analysis procedure described above partially resulted in no suprathreshold clusters for brainstem, midbrain, cortical, and subcortical areas (see Supplementary Results 2 for details).”

3. Similar to point 2 above, there is also a lot of contextualization of findings in the Results that is best left for the Discussion (e.g., p.7 "This is in line with animal studies showing..."; p. 7 "This dovetails findings from electrophysiological.."; p.9 "In line with this, animal studies show that NE release.."; other examples throughout). Since the authors chose to separate Results and Discussion sections, it would benefit the reader to streamline the paper and simply present results in the Results section, and provide interpretation/context in the Discussion, unless they feel this is critical for the paper. I ultimately leave this up to the authors and editor, but combined with my point 2 above, I feel the manuscript would benefit from some streamlining/reorganization to increase readability.

We thank the reviewer for the feedback. We have decided to support the results with short explanations so that a broader readership can better understand the results. Nevertheless, we agree with you that an excessive discussion already in the results section might distract the reader, which is why we have made the following streamlining and adjustments to increase readability of the results:

Therefore, we have removed the following contextualisation of findings in the Results:

~~"[...] While younger adults did not show significant activations in the LC, older adults showed a higher activation of the left LC [T = 4.11, pFWE = 0.04 (voxel cut-off p < 0.005); pFWE = 0.02 (voxel cut-off p < 0.003) (Fig. 2a)] during (1) loss > gain feedback (see Supplementary Table 1). This is in line with animal studies showing that the LC supports the processing of negative events preferentially^{8,33}. Additionally, older adults also showed higher activation of the right LC and right SNr [...]"~~

~~"[...] In line with this, animal studies show that NE release from the LC facilitates memory encoding of negative events^{42,43}. Specifically, older adults showed higher activation of the right LC during encoding of later (3) remembered > not remembered stimuli [...]"~~

~~"[...] including parts of the temporal and parietal lobes, amygdala, and HPC, due to the smaller field of view (FoV) (see Fig. 6). Consistent with previous reports of the middle temporal gyrus (MTG) supporting the processing of negative emotional or salient events^{44,45} and the superior temporal gyrus (STG) supporting information processes⁴⁶ higher activations were generally observed in the MTG as well as STG during salient feedback. Specifically, during (1) loss > gain feedback (see Supplementary Fig. 2; Table 15-17), younger and older adults showed stronger functional activation in the right MTG [...]"~~

Minor

- Top p.4 “cognitively normal younger and older adults” I would avoid using the phrase “cognitively normal” since it implies a more thorough evaluation of cognition/function, whereas these subjects were administered a single test of fluid intelligence. Also, did the authors compare all subjects performance on the test to see if they were within normative expectations? I didn’t see this in the text

We would like to thank the reviewer for the important feedback; indeed, care should be taken when using the term “cognitively normal”. We also adjusted the wording in the manuscript once again and explicitly formulated “fluid intelligence”. The cognitive tests used here were not extensive enough to establish “representativeness” for the older sample, but merely served as a quick check of an indicator of cognitive age differences, showing that older adults have a decline in fluid intelligence compared to younger adults (e.g. Li et al., 2004).

For this reason, we have made the following changes:

“[...] Moreover, by comparing younger and older adults, we investigate (5) age differences in LC reactivity in these instances.”

“[...]An abbreviated version of the Raven's Progressive Matrices⁹⁷ was used to examine whether subjects matched known markers of age differences in adult fluid intelligence⁹⁸[...]”

- Why have the authors chosen to bold certain words or phrases throughout? It seems random and inconsistent (e.g., “left MTG” vs “bilateral MTG”; sometimes brain regions are bolded, sometimes conditions are bolded, sometimes important take home messages are bolded). Minor point but just something to consider.

Thanks for the feedback - the bold letters should draw attention to the essentials. However, as the journal does not allow bold, we have removed bold markings.

- Larger font for activation scale and “T-value” in figures

Thanks for the feedback – we have adjusted the size of the fonts in the respective figures.